# Drug-induced change in transmitter identity is a shared mechanism generating cognitive deficits

Marta Pratelli [1,2] ✉, Anna M. Hakimi [1,2], Arth Thaker[1,2], Hyeonseok Jang [1], Hui-quan Li[1,2], Swetha K. Godavarthi [1,2], Byung Kook Lim [1] & Nicholas C. Spitzer [1,2] ✉

Cognitive deficits are long-lasting consequences of drug use, yet the convergent mechanism by which classes of drugs with different pharmacological properties cause similar deficits is unclear. We find that both phencyclidine and methamphetamine, despite differing in their targets in the brain, cause the same glutamatergic neurons in the medial prefrontal cortex of male mice to gain a GABAergic phenotype and decrease expression of their glutamatergic phenotype. Suppressing drug-induced gain of GABA with RNA-interference prevents appearance of memory deficits. Stimulation of dopaminergic neurons in the ventral tegmental area is necessary and sufficient to produce this gain of GABA. Drug-induced prefrontal hyperactivity drives this change in transmitter identity. Returning prefrontal activity to baseline, chemogenetically or with clozapine, reverses the change in transmitter phenotype and rescues the associated memory deficits. This work reveals a shared and reversible mechanism that regulates the appearance of cognitive deficits upon exposure to different drugs.

Brain impairments are often characterized by constellations of symptoms and behavioral alterations, some of which are shared across disorders. Cognitive deficits are found in mood and neuropsychiatric disorders such as drug misuse, schizophrenia and depression, raising the possibility that shared mechanisms could produce the same impairments in response to different stimuli. While studies have focused on the actions of single drugs, less attention has been given to investigating the mechanisms of action that different drugs of abuse have in common. We investigated the effect of sub-chronic treatment with phencyclidine (PCP) or methamphetamine (METH), two drugs belonging to different classes of chemicals. PCP affects glutamatergic transmission by acting as an NMDA antagonist[1], while METH affects signaling by dopamine and other monoamines[2]. Although differing in their molecular targets in the brain and in some behavioral effects[3–6], PCP and METH have been extensively studied for their ability to cause

long-lasting cognitive deficits and mimic symptoms of schizophrenia[7–9]. However, the process by which they generate similar behavioral impairments has been unknown. Understanding shared neuronal mechanisms underlying drug-induced cognitive deficits could foster development of effective treatments and be beneficial for a spectrum of disorders[10,11].

When neuronal activity is altered for a sustained period, neurons can change the neurotransmitter they express, often switching from an excitatory to an inhibitory transmitter or vice versa and causing changes in behavior[12,13]. Using a combination of genetic labeling strategies, RNA interference, chemogenetics and optogenetics, we investigated whether changes in cortical neuron transmitter phenotype are involved in the generation of both PCP- and METH-induced cognitive deficits.

Here we show that exposure to either PCP or METH induces the same glutamatergic neurons in the prelimbic subregion (PL) of the

[1]Neurobiology Department, School of Biological Sciences and Center for Neural Circuits and Behavior, University of California San Diego, La Jolla, CA 92093-0955, USA. [2]Kavli Institute for Brain and Mind, University of California San Diego, La Jolla, CA 92093-0955, USA. ✉e-mail: mpratelli@ucsd.edu; nspitzer@ucsd.edu

medial prefrontal cortex (mPFC) to gain a GABAergic phenotype, and that overriding this gain of GABA is sufficient to prevent the appearance of drug-induced cognitive deficits. We also demonstrate that both hyperactivity of the PL and increased release of dopamine from VTA neurons mediate this drug-induced change in transmitter phenotype and can be leveraged to prevent or reverse it.

## Results

### PCP changes the transmitter of PL neurons causing cognitive deficits

To determine whether changes in neurotransmitter phenotype are involved in generating cognitive deficits, we tested the effect of a 10-day treatment with PCP (10 mg/kg/day) that induces long-lasting cognitive impairments and recapitulates deficits observed in schizophrenia[8,14]. We focused on the medial prefrontal cortex (mPFC), which is a major hub for cognitive control[15,16], and we examined the transmitter phenotype of neurons expressing the vesicular glutamate transporter 1 (VGLUT1) that are the largest neuronal population in the mPFC[16]. To identify these neurons following changes in transmitter profile, we labeled them permanently with a nuclear mCherry reporter using VGLUT1^CRE::mCherry mice (Fig. 1a).

In mice exposed to PCP, we identified 1198 ± 59 mCherry^+ neurons in the PL of the mPFC immunolabelled for GABA and 1096 ± 81 labeled for its synthetic enzyme, glutamic acid decarboxylase 67 (GAD67) (Fig. 1b–d and Supplementary Fig. 1a–c). In control mice, there were only 622 ± 45 and 643 ± 22 of these neurons, indicating that PCP increased the number GABA^+/mCherry^+ and GAD67^+/mCherry^+ cells by 1.9- and 1.7-fold. PCP did not alter the number of mCherry-labeled PL neurons (Supplementary Fig. 1d), and no sign of apoptosis or neurogenesis was detected (Supplementary Fig. 2). These results suggested that PCP induces the synthesis and expression of GABA in PL glutamatergic neurons not previously expressing this transmitter.

To learn whether the observed co-expression of mCherry and GABA is caused by unspecific mCherry expression, we used fluorescent in situ hybridization (FISH) to determine the extent to which mCherry labeling recapitulates VGLUT1 expression in the PL of VGLUT1^CRE::mCherry mice (Supplementary Fig. 3a, b). In both saline- and PCP-treated mice, 96% of neurons expressed both VGLUT1 and mCherry, while 3% expressed VGLUT1 but not mCherry and only -1% expressed mCherry but not VGLUT1 (Supplementary Fig. 3c). To find out whether these VGLUT1^-/mCherry^+ neurons co-expressed GABAergic markers, we quantified mCherry and VGLUT1 transcript colocalization with transcripts for the GABA synthetic enzyme (GAD1). 0.08% of mCherry^+ neurons in saline- and 0.05% in PCP-treated mice were VGLUT1^- and GAD1^+ (Supplementary Fig. 3d). Instead, GAD1 colocalization was most frequently detected in neurons labeled with both VGLUT1 and mCherry (0.8% of total glutamatergic neurons in saline-treated controls, and 2.2% in PCP-treated mice) (Supplementary Fig. 3e). Thus, in both PCP- and saline-treated VGLUT1^CRE::mCherry mice, unspecific expression of mCherry contributes ≤0.08% to the observed co-expression of GABA and mCherry in PL neurons (Fig. 1b, c).

We next examined the expression levels of the GABA vesicular transporter (VGAT) and VGLUT1 in mCherry^+ neurons that gained GABA after PCP treatment or that co-expressed GABA in drug-naïve conditions, and compared those levels with the expression levels of VGAT and VGLUT1 in neurons expressing only GABA or only glutamate. We used FISH to detect transcripts for mCherry, GAD1, and for either VGAT or VGLUT1. To reveal changes in expression levels, we selectively decreased the amplification for VGAT and VGLUT1 to obtain punctate staining (Fig. 1e, f). In the PL of PCP-treated mice, neurons expressing both mCherry and GAD1 (GAD1^+/mCherry^+) expressed VGAT at the level of GABAergic neurons (labeled with GAD1 and not mCherry) (Fig. 1g). At the same time, the expression level of VGLUT1 in GAD1^+/mCherry^+ neurons decreased by -55% compared to that of

glutamatergic cells expressing mCherry and not GAD1 in PCP-treated mice (Fig. 1h). Neurons co-expressing mCherry and GAD1 in drug-naïve conditions also showed high expression levels of VGAT and low VGLUT1 (46% less VGLUT1 than in purely glutamatergic neurons expressing only mCherry), as evident from analyses of these cells in saline-treated controls (Supplementary Fig. 4a–c). The expression level of VGAT and VGLUT1 in GAD1^+/mCherry^+ neurons did not differ between PCP- and saline-treated mice (Supplementary Fig. 4d, e), but the number of GAD1^+/mCherry^+ neurons was higher in the PL of PCP-treated mice than in controls (6.6 ± 0.5 neurons/mm², saline; 10.6 ± 0.9 neurons/mm², PCP), mirroring the PCP-induced increase in the number of GAD67^+/mCherry^+ neurons (Fig. 1c) and suggesting that PCP induced gain of GAD1 in mCherry^+ neurons that were not expressing it earlier. Thus, both the glutamatergic neurons that gained GABA after PCP-exposure, as well as those expressing GABA in drug-naïve conditions, express high levels of VGAT and lowered levels of VGLUT1.

We then asked whether PCP-treatment affects the transmitter phenotype of PL GABAergic neurons. No difference was observed in the number of neurons expressing GABA and not mCherry between PCP-treated animals and controls (8594 ± 340 vs 8837 ± 271) (Supplementary Fig. 1e). However, PCP and other NMDA receptor antagonists have been shown to reduce the expression of GAD67 and parvalbumin in prefrontal cortex parvalbumin-positive (PV^+) interneurons[17,18]. Because variability in the number of GABAergic neurons scored could have prevented the detection of loss of GABA from a small number of PV^+ neurons, we quantified the number of PV^+ neurons expressing GAD67 after PCP-treatment by permanently labeling them with a PV^CRE::TdTomato mouse line. PCP caused 223 ± 45 TdTomato^+ neurons (6% of the TdTomato^+ population) to stop expressing GAD67 (Supplementary Fig. 5), confirming that PCP treatment reduces the expression of GABAergic markers in a subpopulation of PV^+ neurons[18].

We next asked whether neurons that change their transmitter phenotype in response to PCP-treatment play a role in the appearance of drug-induced cognitive deficits. To investigate whether glutamatergic neurons that have gained GABA contribute to cognitive deficits, we selectively suppressed GABA expression in PL glutamatergic neurons by injecting a Cre-dependent adeno-associated virus (AAV) expressing shRNA for GAD1 (AAV-DIO-shGAD1-GFP or AAV-DIO-shScr-GFP as control) in the PL of VGLUT1^CRE mice before exposure to PCP (Fig. 1i, j). shGAD1 suppressed GABA expression in neurons expressing the virus (Supplementary Fig. 6a–c), and reduced the number of PL neurons co-expressing GAD1 and VGLUT1 transcripts in both PCP- and saline-treated mice to half of that in saline-ShScr controls (Fig. 1k). Having efficiently suppressed PCP-induced gain of GABA, we examined the impact of shGAD1 on PCP-induced behavior. While shGAD1 did not affect PCP-induced hyperlocomotion on the first day of treatment, it prevented appearance of locomotor sensitization after a 10-day PCP-exposure (Fig. 1l and Supplementary Fig. 6d), indicating that PCP-induced gain of GABA is required for sensitization to the acute locomotor effect of the drug. We next focused on deficits in recognition and working memory, since these behaviors are affected by repeated exposure to PCP[14,19] and are regulated by the PL[20,21]. shGAD1 prevented PCP-induced impairments of recognition memory in the novel object recognition test (NORT) (Fig. 1m) and deficits in spatial working memory in the spontaneous alternation task (SAT) (Fig. 1n). Neither PCP-treatment nor shGAD1 changed exploratory behaviors in the NORT and in the SAT (Supplementary Fig. 6e, f). shGAD1 did not affect the behavioral performances of saline-treated controls (Fig. 1l–n), suggesting that glutamatergic neurons gaining GABA upon PCP-exposure, but not those co-expressing GABA before drug-exposure, mediated drug-induced locomotor sensitization and memory impairments. The number of PL GAD1^+/VGLUT1^+ neurons was positively correlated with locomotor sensitization, and negatively correlated with object recognition and working memory performance (Fig. 1o–q). Overall, these data indicate that PCP-induced gain of GABA in PL

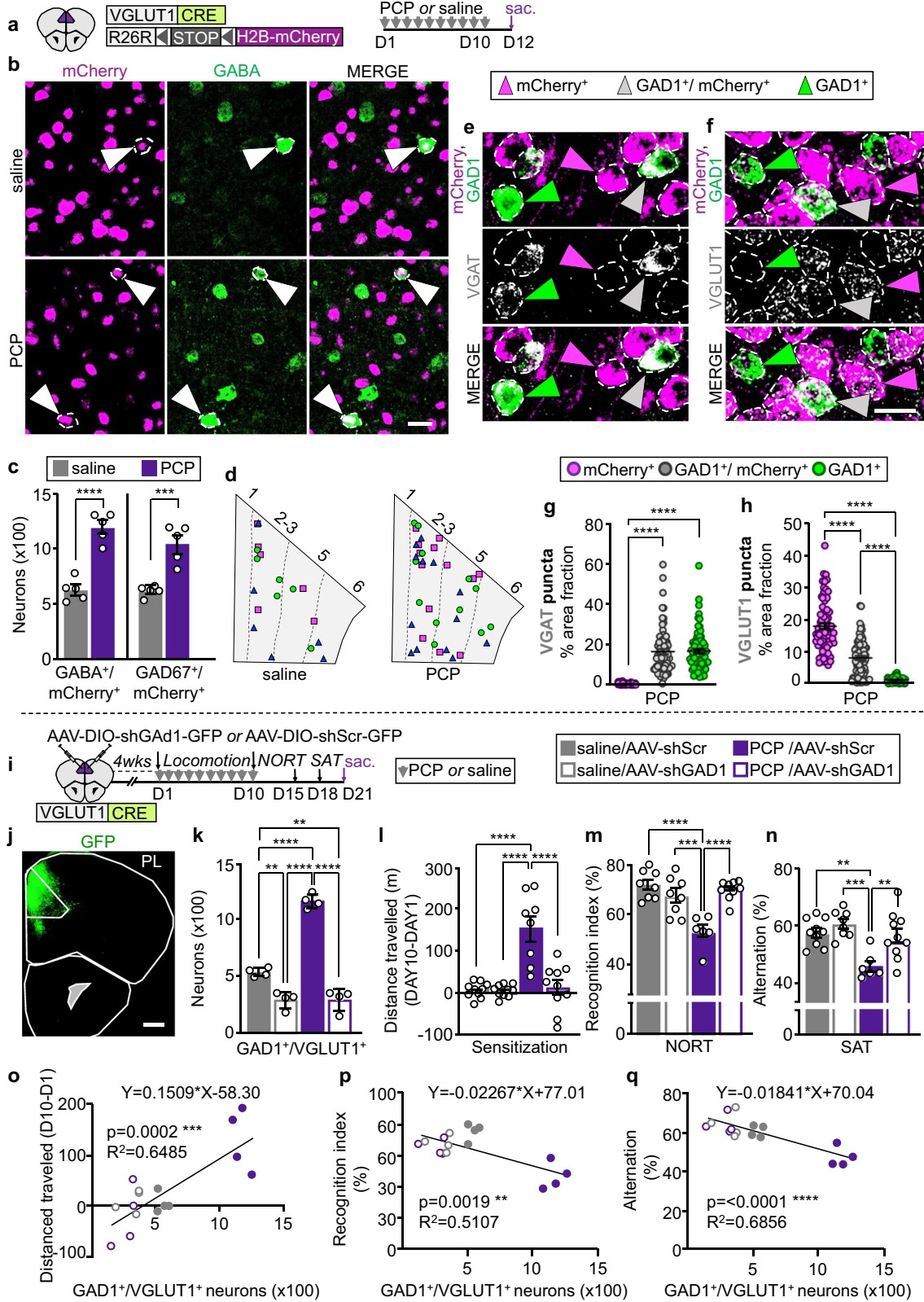

glutamatergic neurons is necessary for the appearance of these behavioral alterations.

It remained unclear whether the loss of GAD67 from PV interneurons was also involved in generating PCP-induced cognitive deficits. To investigate if this was the case, we tested the behavioral effect of suppressing GAD67 expression in PV neurons of the PL in absence of PCP-treatment. PV^CRE mice were injected in the PL with AAV-DIO-

shGAD1-GFP, or AAV-DIO-shScr-GFP as control (Supplementary Fig. 7a). This approach produced an 88% decrease in the number of GFP⁺ PV neurons expressing GABA in mice injected with shGAD1 as compared to controls (Supplementary Fig. 7b, c). However, the behavior of shGAD1-injected mice did not differ from that of control, suggesting that loss of GABA by PL PV⁺ neurons is not sufficient to induce cognitive deficits (Supplementary Fig. 7d–g).

**Fig. 1 | PCP induces GABA expression in PL glutamatergic neurons and suppressing this GABA expression prevents PCP-induced locomotor sensitization and cognitive deficits. a** Experimental protocol. Cartoon after[75]. **b** PL neurons co-expressing mCherry and GABA (arrowheads). Scale bar, 25 µm. **c** Quantification of GABA[+]/mCherry[+] and GAD67[+]/mCherry[+] neurons ($n = 5$ mice). **d** PL locations of glutamatergic neurons that co-expressed GAD1 or changed transmitter identity upon treatment with PCP. Cartoons were produced by superimposing the locations of neurons in three PL sections, each from a different mouse, all positioned at Bregma 1.94. Different shapes represent cells from different mice. **e, f** Expression of VGAT and VGLUT1 mRNA puncta in PL neurons of PCP-treated mice. Scale bar, 20 µm. **g, h** Quantification of VGAT and VGLUT1 expression measured as percent of cell area occupied by mRNA puncta across PL neuronal subtypes ($n = 25$ cells/type/mouse for 3 mice). **i** Experimental protocol. Cartoon after[75]. **j** Expression of shGAD1-GFP in the PL. Scale bar, 500 µm. **k** Quantification of neurons co-expressing VGLUT1 and GAD1 transcripts in AAV-injected mice ($n = 4$ mice). **l–n** shGAD1 prevents PCP-induced deficits in locomotor sensitization (**l**, $n=$ from left to right 10, 10, 8, 10 mice), in the Novel Object Recognition Test (NORT) (**m**, $n=$ from left to right 8, 8, 6, 9 mice) and Spontaneous Alternation Task (SAT) (**n**, $n=$ from left to right 10, 8, 6, 10 mice). **o–q** The number of PL GAD1[+]/VGLUT1[+] neurons is positively correlated with locomotor sensitization and negatively correlated with performance in the NORT and SAT ($n = 4$ mice). Statistical significance (**$P < 0.01$, ***$P < 0.001$, ****$P < 0.0001$) was assessed using two-sided unpaired t-test (**c**), Kruskal–Wallis followed by Dunn's test (**g, h**), two-way ANOVA followed by Tukey's test (**k–n**), and two-sided linear regression and Pearson's correlation analysis (**o–q**). Data are presented as mean ± SEM. The exact p-values and additional statistical details can be found in Supplementary Table 1.

## METH changes the transmitter of the same PL neurons affected by PCP

Because METH causes memory deficits similar to those induced by PCP[21–23], we asked whether METH-treatment would also affect the transmitter identity of PL glutamatergic neurons. Resembling the effect of PCP, 10 days of METH-treatment (1 mg/kg/day) increased the number of mCherry[+] PL neurons co-expressing GABA and GAD67 by 1.7- and 1.9-fold (Fig. 2a–d and Supplementary Fig. 8a), without changing the number of mCherry[+] and GABA[+]/mCherry[-] cells (Supplementary Fig. 8b) or inducing apoptosis or neurogenesis (Supplementary Fig. 2). As previously observed for PCP- and saline-treated animals, unspecific expression of mCherry could not account for the observed co-expression of GABA and mCherry in the PL of mice that received METH (Supplementary Fig. 3). We next used FISH to examine the expression of the VGAT and VGLUT1 in mCherry[+] neurons that co-expressed or gained GABA. Similar to the effects of PCP-treatment, GAD1[+]/mCherry[+] neurons in the PL of METH-treated mice expressed a level of VGAT equal to that of neurons expressing GAD1 but not mCherry (Fig. 2e). The expression level of VGLUT1 decreased by ~76% compared to that of glutamatergic cells expressing mCherry and not GAD1 (Fig. 2f). VGAT and VGLUT1 expression levels in GAD1[+]/mCherry[+] neurons did not differ from those measured in PCP- and saline-treated mice (Supplementary Fig. 4d, e), and the number of GAD1[+]/mCherry[+] neurons in the PL of mice that received METH resembled the number observed in PCP-treated mice ($10.6 ± 0.9$ neurons/mm$^2$, PCP; $10.2 ± 0.5$ neurons/mm$^2$, METH), indicating that both drugs affect PL glutamatergic neurons similarly.

Glutamatergic neurons that co-express GABA or gain it after treatment with either drug were most prevalent in layer 2/3 and layer 5 of the PL (Supplementary Fig. 9a). These PL layers innervate the nucleus accumbens (NAc)[24], which modulates behaviors that are affected by repeated intake of PCP or METH[25–27]. To determine whether neurons that change transmitter identity project to the NAc, we injected fluoro-gold (FG) into the NAc of VGLUT1[CRE]::mCherry mice, treated them with PCP, METH or saline, and screened the PL for GABA[+]/mCherry[+] neurons expressing the retrograde tracer (Supplementary Fig. 9b–d). In both PCP- and METH-treated mice, ~0.9% of FG[+] neurons were GABA[+]/mCherry[+]. Such cells were less frequent in controls (~0.3% of the total number of FG[+] neurons) (Supplementary Fig. 9e), indicating that neurons changing transmitter identity with drug-treatment project to the NAc.

Since both PCP and METH affect the transmitter phenotype of PL glutamatergic neurons that have the NAc as a shared downstream target, we asked whether both drugs change the transmitter identity of the same cells. If PCP and METH changed the transmitter identity of different cells, administering the two drugs one after the other should induce gain of GABA in neurons that have not gained it after treatment with the first drug. To determine whether this was the case, we genetically labeled neurons expressing GABAergic markers during the interval between the delivery of PCP and METH, using VGAT[FLP]::CreER[T]::TdTomato[cON/fON] mice (see Methods) in which

neurons expressing VGAT at the time of tamoxifen administration are permanently labeled with TdTomato (Fig. 3a, b, Supplementary Fig. 10). We first injected tamoxifen in saline-treated controls and determined that TdTomato labels neurons co-expressing VGLUT1 and GAD1 in drug-naïve conditions with 77% efficiency and 79% specificity (Supplementary Fig. 10d, e).

We then used this labeling approach to distinguish neurons expressing GAD1 in drug-naïve mice from those gaining it upon drug-exposure, by administering mice with PCP after saline- and tamoxifen-treatment (Fig. 3c). PCP administration increased the total number of PL GAD1[+]/VGLUT1[+] neurons 2-fold compared to controls ($1188 ± 23$, saline+PCP; $582 ± 27$, saline+saline) (Fig. 3d), in line with previous findings (Fig. 1k). We detected no differences in the number of GAD1[+]/VGLUT1[+]/TdTomato[+] neurons ($441 ± 43$, saline+PCP; $447 ± 34$, saline+saline) and VGLUT1[+]/TdTomato[+] neurons ($99 ± 61$, saline+PCP; $120 ± 8$, saline+saline) between saline+PCP mice and saline+saline controls (Fig. 3d). Changes in these numbers would have indicated that drug-treatment caused some glutamatergic neurons co-expressing GAD1 in drug-naïve conditions to lose expression of GAD1. The results indicate that PCP induces expression of GAD1 in a population of PL neurons that were not previously expressing it, without affecting the transmitter phenotype of cells co-expressing GAD1 and VGLUT1 in drug-naïve conditions.

We next used VGAT[FLP]::CreER[T]::TdTomato[cON/fON] mice to determine if PCP and METH cause the same neurons to change transmitter phenotype. Mice were treated first with PCP followed by tamoxifen administration, and then treated with either saline, PCP or METH (Fig. 3e). Across treatment groups the total number GAD1[+]/VGLUT1[+] neurons was unchanged ($1169 ± 46$, PCP+saline; $1273 ± 69$, PCP + PCP; $1177 ± 45$, PCP + METH) (Fig. 3f), indicating that consecutive administration of drugs does not cause additional glutamatergic neurons to gain GAD1. Furthermore, we did not detect differences in the number of GAD1[+]/VGLUT1[+]/TdTomato[+] neurons ($861 ± 73$, PCP+saline; $831 ± 43$, PCP + PCP; $832 ± 38$, PCP + METH) nor in the number of VGLUT1[+]/TdTomato[+] neurons ($180 ± 44$, PCP+saline; $108 ± 41$, PCP + PCP; $237 ± 103$, PCP + METH) (Fig. 3f). A decrease in the first population and an increase in the second population would have indicated loss of GAD1 from some VGLUT1[+] neurons and gain of GAD1 in another population of VGLUT1[+] neurons. These results indicated that consecutive administration of PCP and METH does not cause gain of GAD1 by additional neurons, nor induces neurons that gained GAD1 upon PCP-treatment to revert to their original transmitter phenotype. Thus, PCP and METH change the transmitter identity of a largely overlapping population of PL neurons.

## Dopamine signaling mediates the change in transmitter phenotype

Demonstration that both PCP and METH have the same effect on the transmitter phenotype of the same PL glutamatergic neurons prompted investigation of shared factors that could mediate this change. PCP, METH, and other addictive substances affect the firing of

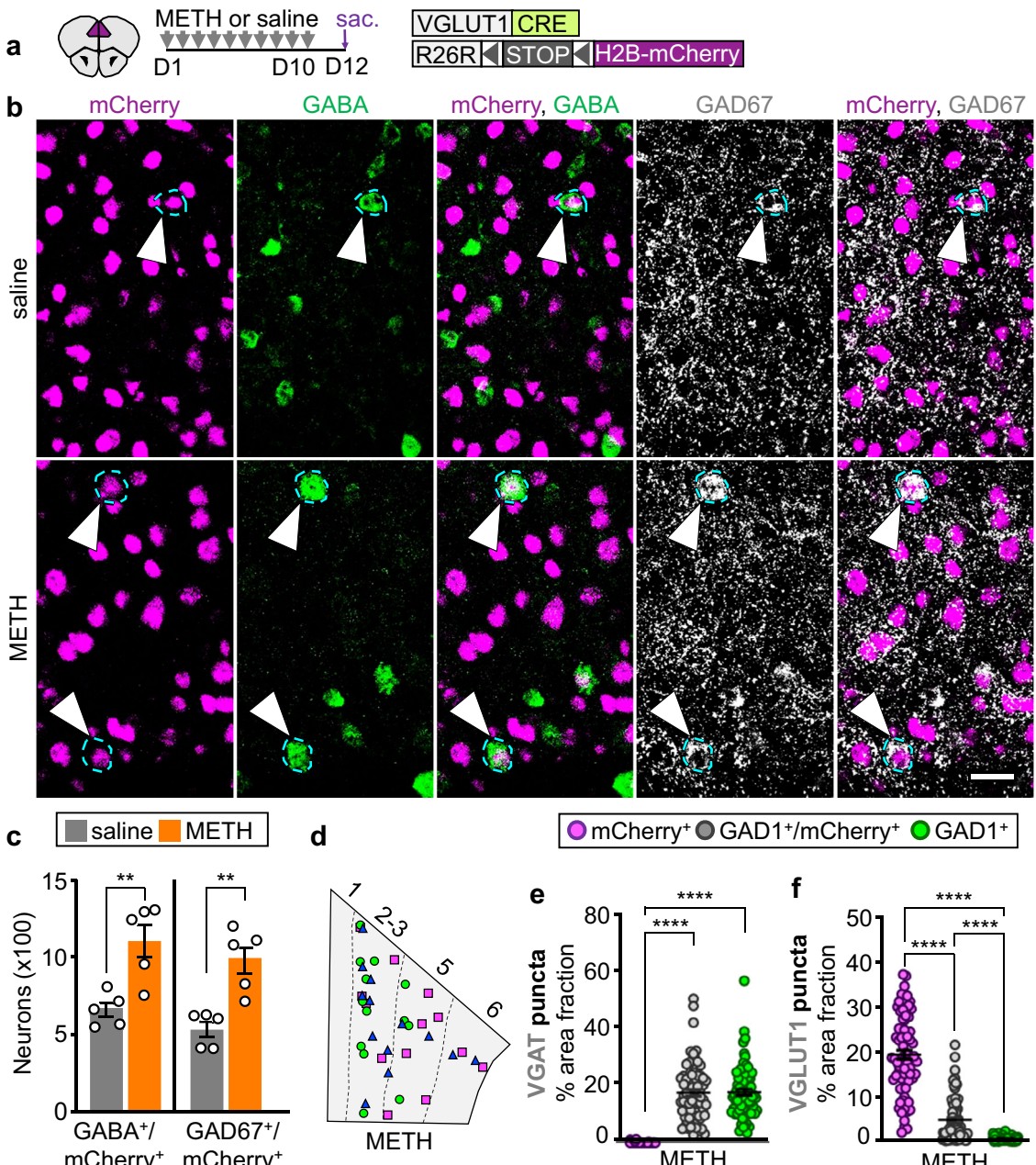

**Fig. 2 | METH causes the same change in PL glutamatergic neuron transmitter phenotype. a** Experimental protocol. Cartoon after[75]. **b** PL neurons co-expressing mCherry, GABA, and GAD67 (arrowheads). Scale bar, 25 µm. **c** Quantification of PL neurons co-expressing mCherry and GABA or GAD67 (n = 5 mice). **d** PL locations of glutamatergic neurons that co-expressed GAD1 or changed transmitter identity upon treatment with METH generated as in Fig. 1d. **e**, **f** VGAT and VGLUT1 expression measured as percent of cell area occupied by mRNA puncta (n = 25 cells/type/mouse for 3 mice). Statistical significance (**P < 0.01, ****P < 0.0001) was assessed using two-sided unpaired t-test and Mann–Whitney U (**c**), Kruskal–Wallis followed by Dunn's test (**e**, **f**). Data are presented as mean ± SEM. The exact p-values and additional statistical details can be found in Supplementary Table 2.

dopaminergic neurons in the ventral tegmental area (VTA)[28,29] and increase the levels of extracellular dopamine (DA) in the mesocorti-colimbic system[30,31]. Could signaling by dopaminergic neurons in the VTA be a common mediator of the PCP- and METH-induced change in transmitter identity? To address this question, we tested whether suppressing the activity of VTA dopaminergic neurons during treatment with PCP or METH affects the number of PL neurons that change transmitter phenotype. We expressed the PSAML-GlyR chemogenetic receptor in the VTA of DAT[CRE] mice (Supplementary Fig. 11a–d). Administration of the PSEM[308] ligand before drug-injection suppressed the acute PCP- and METH-induced increase in c-fos[+] dopaminergic neurons (Supplementary Fig. 11e, f). Combining VTA suppression with

drug administration for the entire duration of treatment, by co-administration of PSEM[308] and PCP or METH, prevented the increase in the number of PL GAD1[+]/VGLUT1[+] neurons (Fig. 4a–d). These results show that drug-induced increase in activity of VTA dopaminergic neurons is required for PL neurons to change their transmitter phenotype upon treatment with PCP or METH.

It remained unknown, however, whether signaling by dopaminergic neurons in the VTA is by itself sufficient to induce PL neurons to change transmitter identity, or whether other effects of PCP or METH are involved. Optogenetic stimulation of VTA dopaminergic neurons to produce phasic firing can mimic the increase in the levels of mesolimbic dopamine induced by the intake of addictive drugs[32,33]. To

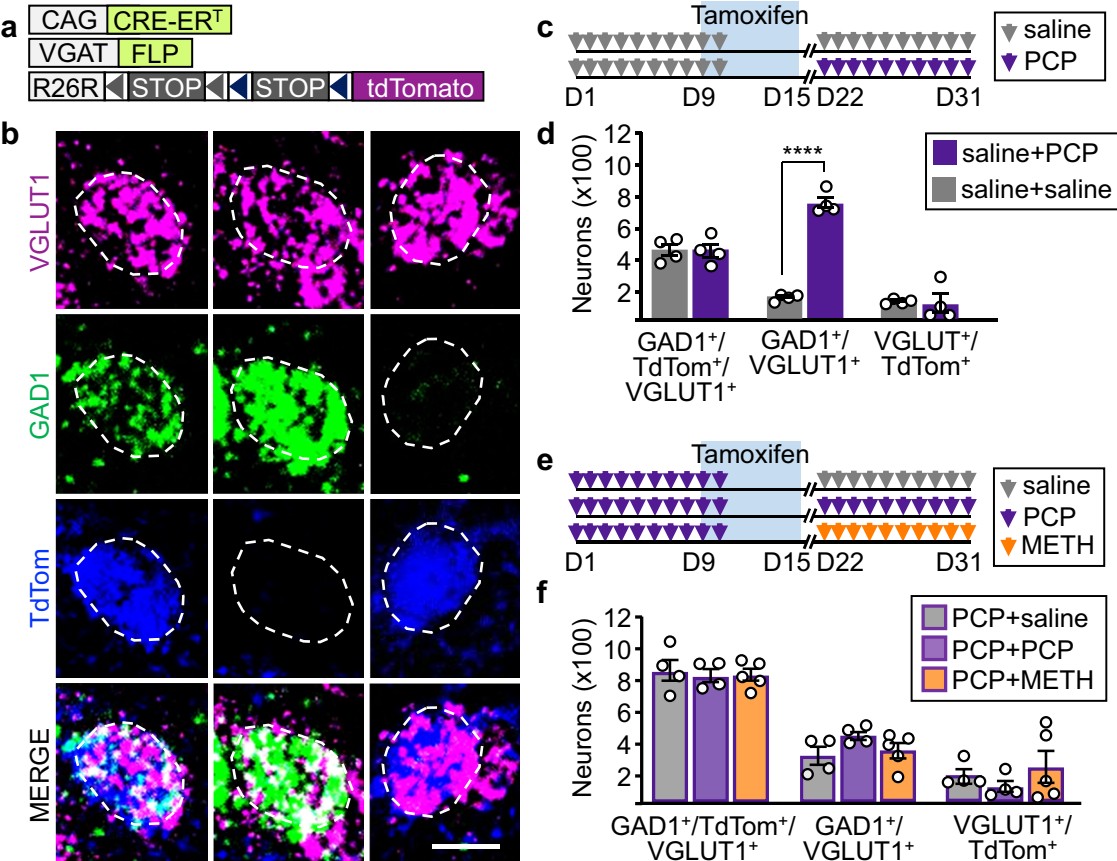

**Fig. 3 | PCP and METH induce gain of GAD1 in the same PL glutamatergic neurons. a** Mouse line used for tamoxifen-inducible genetic labeling. **b** Examples of neurons that are GAD1⁺/VGLUT1⁺/TdTom⁺, GAD1⁺/VGLUT1⁺, and VGLUT1⁺/TdTom⁺. Scale bar, 10 μm. **c** Experimental protocol to distinguish VGLUT1 neurons co-expressing GAD1 in drug-naïve conditions from those gaining GAD1 upon PCP treatment. **d** Quantification of the neurons shown in **b** in mice treated as described in **c** (n = 4 mice). **e** Experimental protocol to learn whether serial administration of

PCP and METH changes the transmitter phenotype of the same number of neurons as PCP alone, causes neurons that have gained GAD1 to lose it, or enables other neurons to gain GAD1. **f** Quantification of neurons shown in **b** in mice treated as described in **e** (n = 4 PCP + saline, 4 PCP + PCP, 5 PCP + METH-treated mice). Statistical significance (****$P < 0.0001$) was assessed using and two-way ANOVA followed by Tukey's test. Data are presented as mean ± SEM. The exact $p$-values and additional statistical details can be found in Supplementary Table 3.

learn whether repeated optogenetic stimulation of VTA dopaminergic neurons is sufficient to induce PL glutamatergic neurons to change transmitter identity, we expressed ChR2-YFP (or YFP as control) in VTA DAT^CRE neurons and implanted an optic fiber above the VTA (Fig. 4e, f). Laser stimulation (each set consisting of 30 bursts at 4 Hz of 5 pulses of 4 ms duration at 20 Hz) of acute brain slices evoked phasic firing of ChR2⁺ neurons (Supplementary Fig. 12a, b). In vivo administration of 80 of these sets of stimulation over the course of 1 h increased the number of VTA dopaminergic neurons expressing c-fos by 7.3-fold in mice expressing ChR2 (Supplementary Fig. 12c–f). We then exposed mice to 1 h of VTA stimulation per day for 10 days and analyzed the transmitter phenotype of PL glutamatergic neurons. The number of GAD1⁺/VGLUT1⁺ neurons was 1.7-fold higher in ChR2-expressing mice compared to controls (Fig. 4g, h, Supplementary Fig. 12j–l), demonstrating that phasic firing of dopaminergic neurons in the VTA is sufficient to change the transmitter phenotype of PL glutamatergic neurons.

We next asked whether direct dopamine signaling in the PL is sufficient to change the transmitter phenotype, or if dopamine signaling affecting the PL indirectly via multi-synaptic pathways is required. We reasoned that if dopamine signaling in the PL were sufficient to induce the change in transmitter phenotype, then this change would be observed after selective stimulation of dopaminergic projections from the VTA to the PL. We injected ChR2-YFP (or YFP as a control) into the VTA of DAT^CRE mice and implanted optic fibers

bilaterally above the PL (Supplementary Fig. 13a–c). We stimulated ChR2⁺ fibers in the PL for one hour a day for 10 consecutive days using the same stimulation parameters as before. However, the number of GAD1⁺/VGLUT1⁺ neurons in the PL of ChR2-expressing mice remained unchanged compared to YFP-expressing controls (Supplementary Fig. 13d). This finding suggests that the change in the transmitter phenotype of neurons in the PL requires increased dopamine signaling in regions other than the PL, and that optogenetic stimulation of dopaminergic cell bodies in the VTA (Fig. 4e, f) induces this change, at least in part, via multi-synaptic pathways.

Overall, these results establish signaling by dopaminergic neurons in the VTA as a common mediator for PCP- and METH-induced gain of GABA in PL glutamatergic neurons and suggest that exposure to other addictive substances that activate the VTA could produce similar effects.

## PL hyperactivity mediates the change in transmitter and behavior

We then asked how the effects of three different stimuli, PCP, METH and stimulation of dopaminergic neurons in the VTA, converge to change the transmitter identity of glutamatergic neurons in the PL. Increased neuronal activity can cause neurons to change the transmitter they express[12,34,35]. Could PCP, METH and optogenetic VTA stimulation induce alterations in PL activity that mediate the change in PL neuron transmitter phenotype? PCP and METH increased c-fos

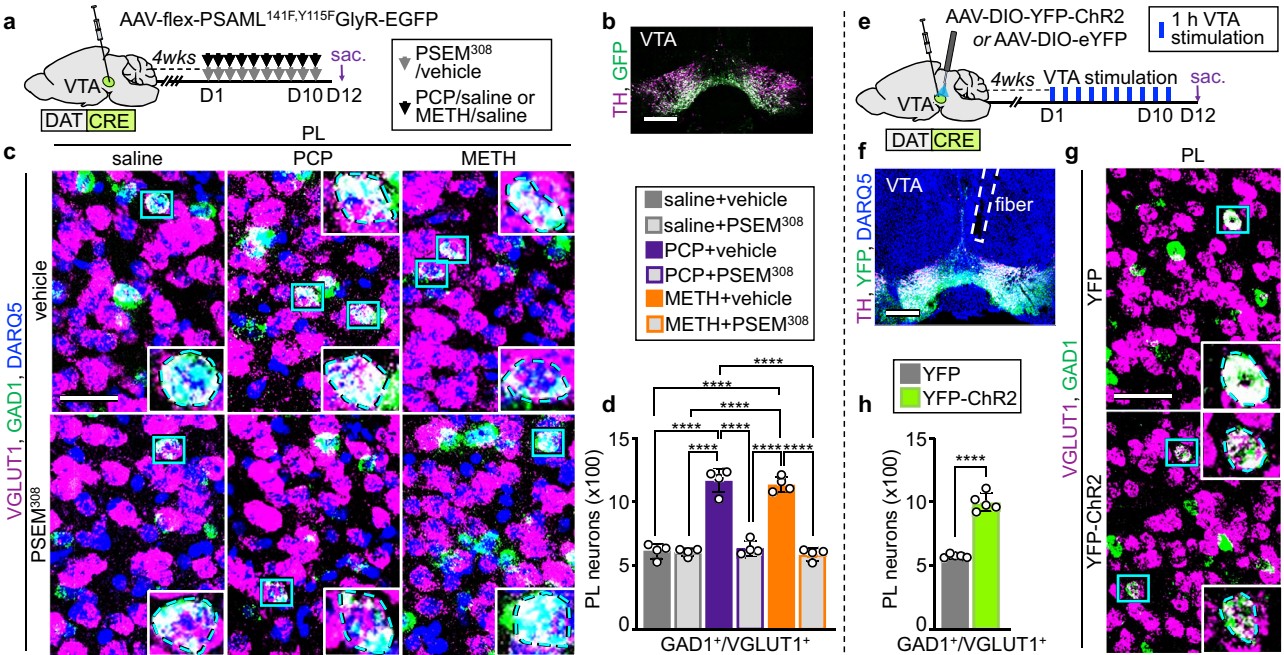

**Fig. 4 | Activity of VTA DA neurons is necessary and sufficient to induce PL glutamatergic neurons to switch their transmitter identity. a** Experimental protocol. Cartoon after[75]. **b** Expression of PSAML-GlyR-GFP in the VTA. Scale bar, 500 μm. **c** VGLUT1 and GAD1 expression in the PL following saline/drug treatment combined with chemogenetic inhibition of VTA dopaminergic neurons. (Blue rectangles) neurons co-expressing VGLUT1 and GAD1 illustrated at high magnification in the insets. Scale bar, 50 μm. **d** Quantification of GAD1$^+$/VGLUT1$^+$ neurons in the PL ($n = 4$ mice). **e** Experimental protocol. Cartoon after[75]. **f** Expression of ChR2-YFP in

the VTA and fiber location. Scale bar, 500 μm. **g** VGLUT1 and GAD1 expression in the PL following 10 days of VTA optogenetic stimulation. (Insets) Higher magnification of neurons co-expressing VGLUT1 and GAD1. Scale bar, 50 μm. **h** Quantification of GAD1$^+$/VGLUT1$^+$ neurons in the PL ($n = 5$ mice). Statistical significance (****$P < 0.0001$) was assessed using two-way ANOVA with Tukey's multiple-comparisons test (**d**) and two-sided unpaired t-test (**h**). Data are presented as mean ± SEM. The exact $p$-values and additional statistical details can be found in Supplementary Table 4.

expression in PL glutamatergic neurons by 3.8- and 3.5-fold after a single injection and by 2.6- and 3.7-fold throughout a 10-day treatment (Supplementary Fig. 14a–d, f, g). Similarly, expression of c-fos in PL glutamatergic neurons was increased by 2.7-fold in mice expressing ChR2 after 1 h of phasic stimulation of VTA dopaminergic neurons (Supplementary Fig. 12g–i). To determine whether this increase in activity promoted the change in transmitter phenotype, we tested whether suppression of PL hyperactivity during treatment with PCP or METH would prevent glutamatergic neurons from gaining GABA. Glutamatergic cells in the PL receive perisomatic inhibition from local PV$^+$ interneurons, which do not show changes in c-fos expression after administration of PCP or METH (Supplementary Fig. 14e, h). We hypothesized that chemogenetic activation of PV$^+$ neurons would suppress drug-induced hyperactivity of glutamatergic cells[36,37]. To test this idea, we expressed the chemogenetic receptor PSAML-5HT3HC in mPFC PV$^+$ neurons using AAV-flex-PSAML$^{141F,Y115F}$5HT3HC-GFP and administered the PSEM$^{308}$ ligand immediately before acute injection of either PCP or METH (Supplementary Fig. 15a–e). While GFP$^+$ neurons infected with the virus showed high c-fos expression after PSEM$^{308}$ treatment, consistent with their expected activation (Supplementary Fig. 15f), the PCP- or METH-induced increase in c-fos expression in PL glutamatergic neurons was suppressed (Supplementary Fig. 15g–i).

We subsequently combined chemogenetic activation of PL PV$^+$ interneurons with either PCP- or METH-administration for the duration of drug-treatment (Fig. 5a, b). The number of GAD1$^+$/VGLUT1$^+$ neurons in the PL of PCP- and METH-treated mice that received PSEM$^{308}$ was half of that of mice that did not (586 ± 18 and 611 ± 23 vs 1262 ± 66 and 1222 ± 12) and was indistinguishable from that of saline-treated controls (Fig. 5c, d). Thus, suppression of drug-induced PL hyperactivity is sufficient to prevent glutamatergic neurons from changing their transmitter identity, indicating that hyperactivity is required for the change in transmitter phenotype.

We now tested whether blocking the change in transmitter phenotype through chemogenetic activation of PV$^+$ neurons was sufficient to prevent drug-induced changes in behavior. In mice treated with PSEM$^{308}$, drug-induced hyperlocomotion was absent on both the first and last days of treatment (distance traveled on DAY1: 30 ± 16 m, saline +vehicle; 29 ± 18 m, saline+PSEM$^{308}$; 173 ± 69 m, PCP+vehicle; 45 ± 25 m, PCP + PSEM$^{308}$; 137 ± 37 m, METH-saline; 35 ± 14 m, METH + PSEM$^{308}$. Distance traveled on DAY10: 32 ± 19 m, saline+vehi-cle; 34 ± 30 m, saline+PSEM$^{308}$; 309 ± 67 m, PCP+vehicle; 67 ± 26 m, PCP + PSEM$^{308}$; 192 ± 30 m, METH-saline; 57 ± 18 m, METH + PSEM$^{308}$). These results are consistent with suppression of acute drug-induced hyperactivity of PL glutamatergic neurons[38]. Suppressing PL activity prevented both PCP- and METH-induced appearance of memory deficits in both the NORT and the SAT (Fig. 5e–h), without influencing exploratory behaviors (Supplementary Fig. 15j–n). The number of GAD1$^+$/VGLUT1$^+$ neurons in the PL was negatively correlated with the performance in the NORT and SAT (Fig. 5i–l). These results suggest that chemogenetic activation of PV$^+$ neurons affects the performance of drug-treated mice by preventing the change in the transmitter phenotype of glutamatergic neurons in the PL.

### Leveraging PL activity reverses transmitter and behavior changes

PL glutamatergic neurons that change their transmitter upon exposure to PCP or METH retain the GABAergic phenotype for at least 11 days of drug washout (Fig. 5d, and PCP+saline group in Fig. 3f). As PCP- and METH-induced memory deficits are also long-lasting[14,39], we asked whether the persistence of behavioral deficits is linked to retention of the GABAergic phenotype and whether both are reversible.

Clozapine, an antipsychotic drug that acts as a serotonin and dopamine receptor antagonist, reverses PCP-induced deficits in the NORT[14], leading us to investigate whether it also reverses the change in

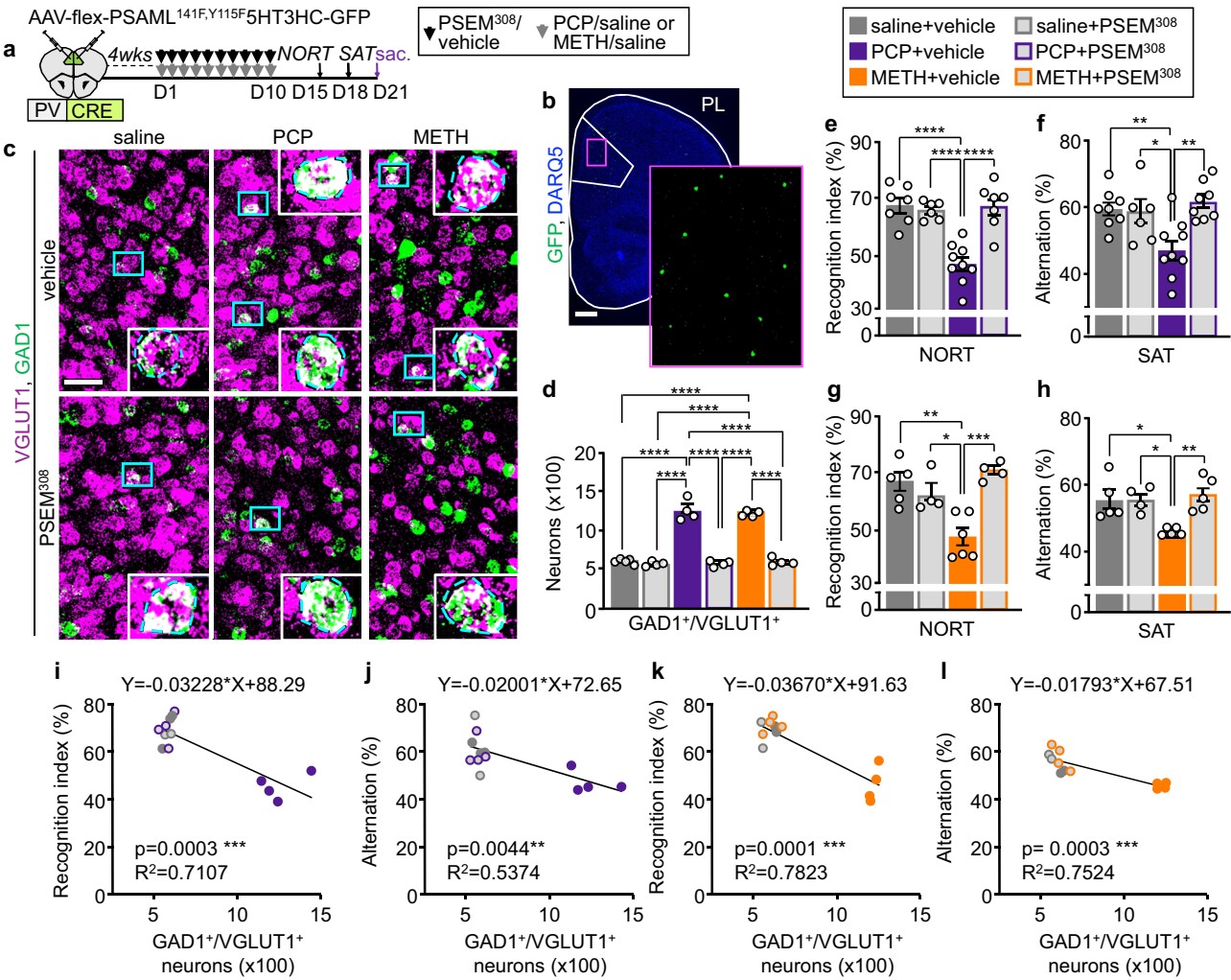

**Fig. 5 | Suppression of drug-induced PL hyperactivity during drug delivery prevents the change in transmitter identity and prevents drug-induced behavioral alterations. a** Experimental protocol. Cartoon after[75]. **b** Expression of PSAML-5HT3HC-GFP in the PL. Scale bar, 500 μm. **c**, VGLUT1 and GAD1 expression in the PL following saline/drug treatment combined with chemogenetic activation of PV+ neurons. (Blue rectangles) neurons co-expressing VGLUT1 and GAD1 illustrated at high magnification in the insets. Scale bar, 50 μm. **d** Quantification of (**c**) (n= from left to right 5, 4, 4, 4, 4 mice). **e–h** Chemogenetic activation of PV+ neurons during drug treatment prevents the deficits in the NORT and SAT induced

by both PCP and METH (**e** n= from left to right 7, 6, 9, 7 mice; **f** n= from left to right 8, 6, 9, 8 mice; **g** n= from left to right 5, 4, 6, 4 mice; **h** n= from left to right 5, 4, 5, 5 mice). **i–l** The number of PL GAD1+/VGLUT1+ neurons is negatively correlated with performance in the NORT and SAT. Statistical significance (*$P < 0.05$, **$P < 0.01$, ***$P < 0.001$, ****$P < 0.0001$) was assessed using two-way ANOVA followed by Tukey's test (d-h) and two-sided linear regression and Pearson's correlation analysis (**i–l**). Data are presented as mean ± SEM. The exact p-values and additional statistical details can be found in Supplementary Table 5.

transmitter phenotype. VGLUT1CRE::mCherry mice that received PCP displayed 1.9 fold more GABA+/mCherry+ PL neurons than controls 17 days after the end of PCP-treatment, indicating that neurons had maintained the acquired GABAergic phenotype (Fig. 6a–c). In mice that received clozapine treatment after PCP, the number of GABA+/mCherry+ neurons was reduced compared to that of mice treated with PCP alone (559 ± 55 vs 1124 ± 94) and was not different from that of saline-treated controls (Fig. 6a–c). Clozapine did not affect the number of GABA+/mCherry+ neurons in saline-treated mice, suggesting that this drug selectively reverses the PCP-induced change in glutamatergic neuron transmitter identity. We found that clozapine rescued PCP-induced memory deficits in the NORT and SAT, without affecting the behavioral performance of controls (Fig. 6d–g and Supplementary Fig. 16a–d).

We next investigated whether changes in neuronal activity could underlie the clozapine-induced reversal of the change in transmitter identity. Because clozapine suppresses the acute PCP-induced increase in PL c-fos expression[40,41], we asked whether reversal of the gain of GABA depends on suppression of neuronal activity. Two days after

drug-treatment, the number of c-fos+ glutamatergic neurons was 2.1- and 2.8-fold higher in PCP- and METH-treated mice compared to controls when evaluated within 5 h from the time of day when they had previously received the drugs (Supplementary Fig. 17a, b, d, e). However, no difference the number of PL c-fos+ neurons was detected 11-to-12 h after the time of day that mice received the drugs (Supplementary Fig. 17c). Thirteen and seventeen days after drug-treatment, c-fos expression in the PL of PCP- and METH-treated mice was still higher than in controls when evaluated within 5 h from the time-of-day when they previously received the drugs (Supplementary Figs. 16e–h, 18a–d). These results suggest the presence of a long-lasting, time-of-day dependent drug-induced hyperactivity. Administration of clozapine after PCP-treatment returned c-fos expression to baseline (Supplementary Fig. 16e–h), suggesting that daily PL hyperactivity during drug-washout is necessary to maintain the newly acquired transmitter phenotype. If this were the case, suppressing PL hyperactivity after the transmitter change has occurred could be expected to reverse the change. To test this hypothesis, we chemogenetically activated PV+ neurons for 10 days to normalize c-fos expression in the PL of PCP- or

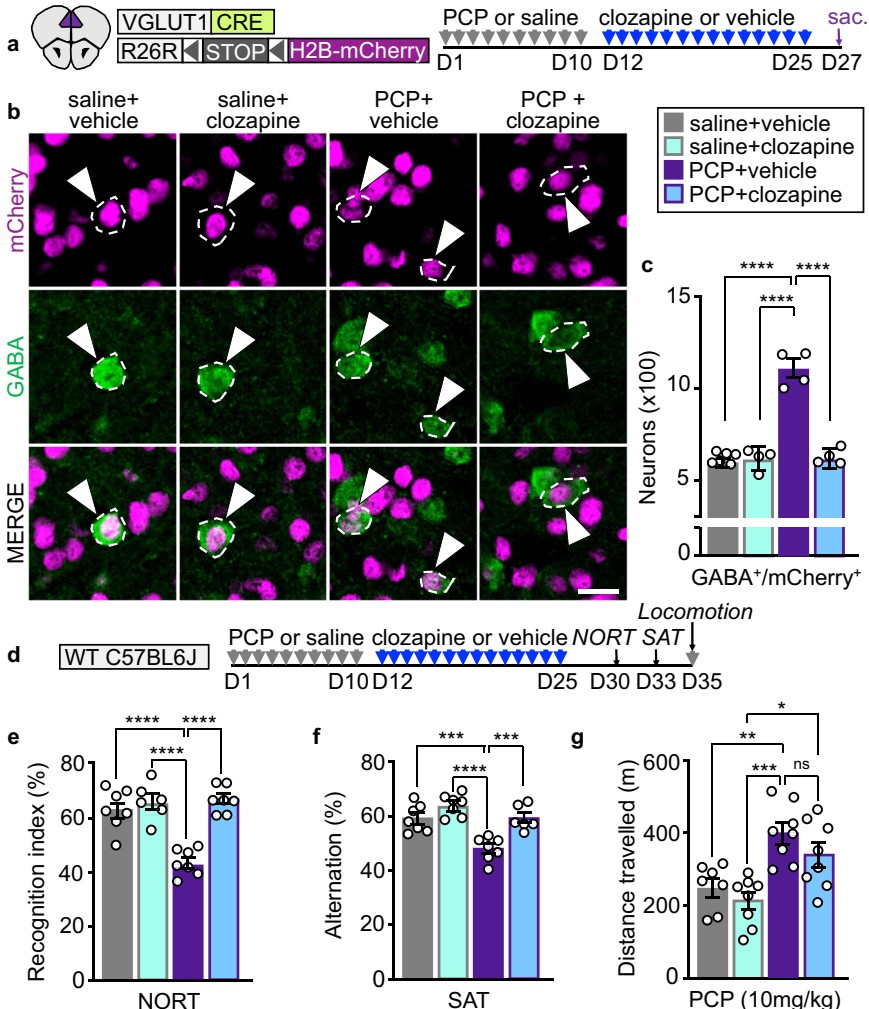

**Fig. 6 | Clozapine treatment reverses PCP-induced gain of GABA in PL neurons and reverses associated behaviors. a** Experimental protocol. Cartoon after[75]. **b** PL neurons co-expressing mCherry and GABA (arrowheads). Scale bar 20 μm. **c** Quantification of PL neurons co-expressing mCherry and GABA after treatment with PCP or saline followed by clozapine or vehicle (*n*= from left to right 6, 4, 4, 4 mice). **d** Experimental protocol. **e**–**g** Clozapine reverses PCP-induced deficits in the

NORT and SAT (**e** *n*= from left to right 7, 6, 7, 7 mice; **f** *n*= from left to right 7, 7, 7, 6 mice) but not locomotor sensitization after a single PCP challenge (**g** *n*= from left to right 7, 8, 8, 8 mice). Statistical significance (**P* < 0.05, ***P* < 0.01, ****P* < 0.001, *****P* < 0.0001) was assessed using two-way ANOVA with Tukey's multiple-comparisons test. Data are presented as mean ± SEM. The exact *p*-values and additional statistical details can be found in Supplementary Table 6.

METH-treated mice after the change in transmitter phenotype had taken place (Supplementary Fig. 18a–d). More than 3 weeks after the end of drug-treatment, glutamatergic neurons in the PL of both PCP and METH-treated mice still displayed the drug-induced GABAergic phenotype (Fig. 7a–c). Normalizing PL activity chemogenetically decreased the number of GAD1[+]/VGLUT1[+] neurons in the PL of PCP and METH-treated mice to the level of controls (588 ± 27 and 575 ± 9 vs 1225 ± 38 and 1071 ± 42) (Fig. 7a–c). Thus, PL neuronal activity maintains the change in transmitter identity once it has been induced. Chemogenetically activating PL PV[+] interneurons after the change in transmitter phenotype had occurred also rescued memory deficits in the NORT and SAT and suppressed locomotor sensitization to both PCP and METH (Fig. 7d–i and Supplementary Fig. 18e–k). Overall, these data show that suppressing PL hyperactivity following drug-exposure reverses the change in transmitter phenotype and the associated behavioral alterations.

## Discussion

We show that gain of a GABAergic phenotype by PL glutamatergic neurons is a shared form of neuroplasticity involved in generating both PCP- and METH-induced cognitive deficits. Both drugs cause the same

PL neurons to acquire a new transmitter phenotype characterized by expression of GABA, GAD67, and VGAT, combined with lower levels of VGLUT1. Other PL neurons show the same transmitter phenotype in drug naïve conditions, as suggested by earlier studies[42,43]. This change in transmitter phenotype regulates the appearance of locomotor sensitization and memory deficits in the NORT and SAT, consistent with the involvement of the PL in the modulation of these behaviors[20,21,25–27,44].

Some of the PL neurons that gain GABA after treatment with PCP or METH project to the NAc. Because glutamatergic inputs from the PL to the NAc are important for reward, drug-seeking and relapse[45–48] the question arises whether gain of GABA in PL-to-NAc neurons influences drug-seeking behaviors. Answering this question will require the use of self-administration paradigms of drug-intake, rather than experimenter-administered drug-delivery, and the demonstration that gain of GABA by PL neurons also occurs in response to drug-self-administration[49].

Signaling by dopaminergic neurons in the VTA is necessary and sufficient to change the transmitter identity of PL neurons. Both PCP and METH increase DA release in the mesocorticolimbic system[30,31]. Methamphetamine may achieve this by directly promoting dopamine

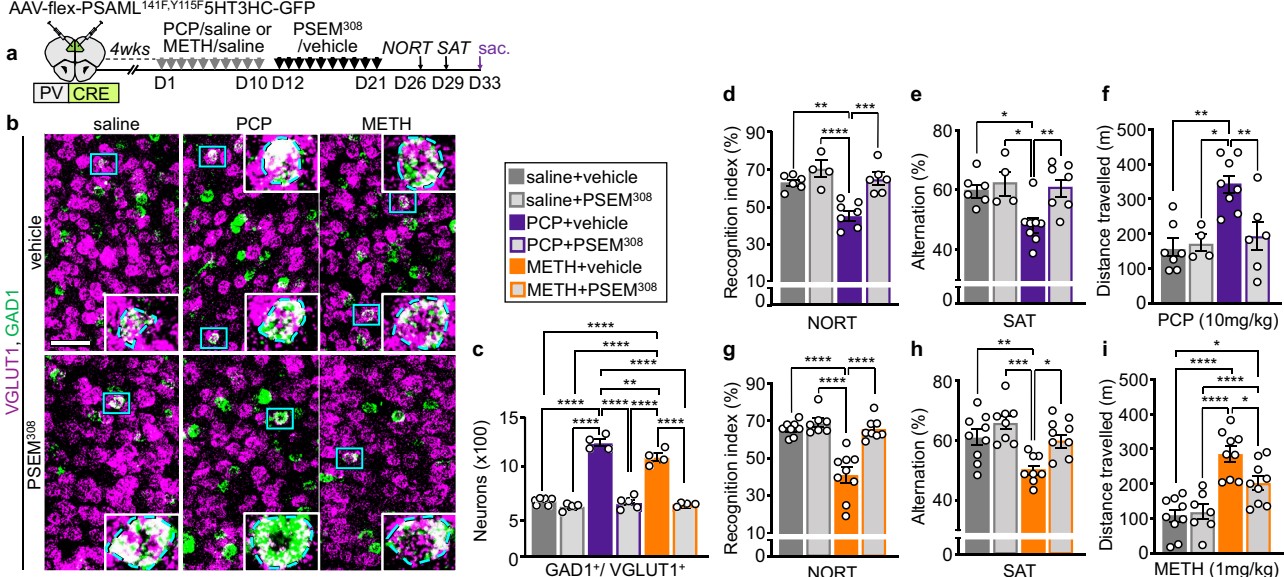

**Fig. 7 | Suppression of drug-induced PL hyperactivity after the end of drug-treatment reverses the change in transmitter identity and reverses drug-induced behavioral alterations. a** Experimental protocol. Cartoon after[75]. **b** PL expression of VGLUT1 and GAD1 after exposure to saline/drug treatment followed by chemogenetic activation of PV+ neurons. (Blue rectangles) higher magnification of GAD1+/VGLUT1+ neurons in insets. Scale bar, 50 μm. **c** Quantification of (**b**) (*n*= from left to right 6, 4, 4, 5, 4, 4 mice). **d**–**i** PCP- and METH-induced locomotor sensitization and deficits in the NORT and SAT are reversed by sustained activation of PV+ neurons (**d** *n*= from left to right 6, 4, 7, 6 mice; **e** *n*= from left to right 6, 4, 8, 7 mice; **f** *n*= from left to right 7, 4, 8, 6 mice; **g** *n*= from left to right 8, 7, 9, 8 mice; **h** *n*= from left to right 9, 8, 8, 9 mice; **i** *n*= from left to right 9, 7, 9, 9 mice). Statistical significance (*$P < 0.05$, **$P < 0.01$, ***$P < 0.001$, ****$P < 0.0001$) was assessed using two-way ANOVA followed by Tukey's test. Data are presented as mean ± SEM. The exact *p*-values and additional statistical details can be found in Supplementary Table 7.

release from dopaminergic neurons[2], while PCP may achieve this by increasing the firing rate of VTA DA neurons[50,51]. Chemogenetic suppression of dopaminergic activity in the VTA during PCP- or METH-treatment prevents the change in transmitter phenotype. Furthermore, optogenetic stimulation of phasic firing of the cell bodies of dopaminergic neurons in the VTA, in the absence of drug administration, is sufficient to induce PL hyperactivity and produce the change in transmitter identity. This result is achieved via multi-synaptic pathways, as selective stimulation of dopaminergic projections from the VTA to the PL is insufficient to induce gain of GABA in glutamatergic neurons of the PL. As glutamatergic neurons in the PL receive inputs from multiple brain regions[52], a more comprehensive investigation will be required to determine the origins of PL inputs directly responsible for the change in transmitter phenotype. Furthermore, VTA dopaminergic neurons can release other transmitters than dopamine[53], which may also contribute to the change in transmitter phenotype of PL neurons. Our results are consistent with evidence that many addictive substances, including PCP and METH, promote phasic firing of dopaminergic neurons in the VTA[54] and that inducing phasic firing of these neurons enhances DA release in the NAc[33].

Neuronal hyperactivity in the PL, as reported by c-fos, mediates the change in transmitter identity, as expected for activity-dependent neurotransmitter switching[12,34,35], and is necessary to maintain the newly acquired transmitter phenotype after the end of drug treatment. Midbrain cholinergic neurons that change transmitter identity in response to sustained exercise spontaneously revert to expression of their original transmitter within a week of cessation of the stimulus[12]. In contrast, PL glutamatergic neurons maintain their GABAergic phenotype for more than 3 weeks after the end of drug-treatment and the linked cognitive deficits are long-lasting[14,39]. c-fos expression in the PL increases after acute treatment with PCP or METH[40,55], as well as after one hour of phasic stimulation of VTA dopaminergic neurons, and remains elevated in a time-of-day dependent manner for at least two weeks[56,57]. Suppressing drug-induced hyperactivity during or after

drug-treatment, respectively, prevents or rescues the change in transmitter phenotype and the change in behavior. While this evidence is not sufficient to determine whether expression of c-fos alone is sufficient to induce the change in transmitter phenotype, it shows that PL hyperactivity, as reported by c-fos, is necessary to produce and maintain this change.

Activity-dependent changes in transmitter phenotype during development can be mediated at the transcriptional level[58,59] and by post-transcriptional mechanisms involving miRNAs[60]. In the former case, neuronal hyperactivity acts in a non-cell-autonomous manner by promoting the release of BDNF, which binds to the TrkB receptor tyrosine kinase on neighboring neurons and initiates a MAP kinase cascade leading to a change in transmitter phenotype mediated by the phosphorylation of the of the c-Jun transcription factor. Because reports of the effect of PCP and METH on BDNF expression in the PL are contradictory[61–65], determining whether similar mechanisms are involved in mediating drug-induced gain of GABA by PL glutamatergic neurons requires additional investigation.

In conclusion we identify a shared and reversible cellular mechanism involved in the ability of different drugs to generate cognitive deficits. Since multiple drugs of abuse acutely promote mPFC hyperactivity[66,67], and stimulation of VTA dopaminergic neurons is sufficient to change the transmitter identity of PL neurons, drugs other than METH and PCP may result in the appearance of cognitive deficits by promoting this change in the transmitter phenotype of PL glutamatergic neurons.

## Methods
### Mice
All animal procedures were carried out in accordance with NIH guidelines and approved by the University of California, San Diego, Institutional Animal Care and Use Committee (protocol: s14016). Mice were maintained on a 12 h:12 h light:dark cycle (light on: 7:00 am–7:00 pm) with ad libitum access to food (7912.15 Irradiated Teklad

LM Mouse/Rat diet) and water. Temperature was maintained between 65 and 75 °F (-18 and 23 °C) with 40–60% humidity. Mice were preferentially housed 3 or 4 per cage with nesting material. After receiving surgery, mice were single-housed with nesting material. All experiments were performed on 8- to 14-week-old male mice.

For collection of brain samples, mice were deeply anesthetized with isoflurane vapor at the sacrifice time point and transcardially perfused with phosphate-buffered saline (PBS) followed by 4% paraformaldehyde (PFA) in PBS. Brains were dissected and post-fixed in 4% PFA overnight (o/n) at 4 °C, before being transferred to 30% sucrose in PBS for 2 days at 4 °C. Mice from which brain samples were not collected, were euthanized by cervical dislocation while deeply anesthetized with $CO_2$.

C57BL/6J mice (JAX#000664) (referred to as "wild-type"), VGLUT1-IRES2-Cre mice (JAX# 037512) (referred to as "VGLUT1^CRE"), Rosa26-LSL-H2B-mCherry (JAX#023139) (referred to as "mCherry"), B6 PV^cre mice (JAX#017320) (referred to as "PV^CRE"), Ai14(RCL-tdT)-D mice (JAX#007914) (referred to as "TdTomato"), Slc32a1-2A-FlpO-D knock-in mice (JAX# 029591) (referred to as "VGAT^FLP"), Ai65(RCFL-tdT) mice (JAX#021875) (referred to as "TdTomato^cON/fON"), CAGGCre-ER™ mice (JAX# 004682) (referred to as "CreER^T") were obtained from Jackson Laboratories. DAT-IRES-Cre mice (JAX#006660) (referred to as "DAT^CRE") were provided by the Davide Dulcis, Thomas Hnasko, and Cory Root laboratories.

Heterozygous VGLUT1^CRE animals were bred with either wild-type mice to obtain VGLUT1^CRE mice, or with homozygous mCherry mice to obtain VGLUT1^CRE::mCherry experimental mice. PV^CRE mice were maintained in homozygosis and bred with either TdTomato homozygous mice to obtain PV^CRE::TdTomato mice or wild-type animals to obtain PV^CRE heterozygous experimental mice.

To obtain VGAT^FLP::CreER^T::TdTomato^cON/fON mice we first bred heterozygous CreER^T mice with either heterozygous TdTomato^cON/fON or heterozygous VGAT^FLP animals to obtain CreER^T::TdTomato^cON/fON and VGAT^FLP::CreER^T breeders. These breeders were then crossed with VGAT^FLP or TdTomato^cON/fON mice, respectively, to obtain VGAT^FLP::CreER^T::TdTomato^cON/fON experimental mice.

Homozygous DAT^CRE mice were bred with wild-type animals to obtain DAT^CRE heterozygous experimental mice.

## Drugs and pharmacological treatments

Phencyclidine hydrochloride (PCP; Sigma-Aldrich, P3029) was dissolved in sterile saline and administered subcutaneously (s.c.) at a dose of 10 mg/kg/day for 10 consecutive days[14]. Methamphetamine hydrochloride (METH; Sigma-Aldrich, M8750) was dissolved in saline and administered s.c. at a dose of 1 mg/kg/day for 10 consecutive days (modified from[21]). To investigate whether drug-treatment affects the transmitter identity of PL neurons, mice were sacrificed either 2 days (Figs. 1c, g, h, 2c, e, f, and 4d, Supplementary Figs. 1c–e, 3–5, 8, and 9e), 11 days (Fig. 1k, Supplementary Fig. 6c, Fig. 5d), 17 days (Fig. 6c) or 23 days (Fig. 7c) after the end of PCP or METH treatment. To investigate the effect of PCP or METH on PL neuronal activity, mice were sacrificed 2 h after a single injection of the drug (Supplementary Fig. 14c–e), 2 h after the last of 10 daily injections (Supplementary Fig. 14f–h) or 2 days after the end of 10 days of drug treatment either within 0–5 h or 11–12 h after the time of daily drug administration (Supplementary Fig. 17).

Tamoxifen (Sigma-Aldrich, T5648) was dissolved in corn oil (Sigma-Aldrich, C8267)/ethanol 9:1 and administered intraperitoneally (i.p.) at a dose of 75 mg/kg/day. To test whether PCP and METH affect the neurotransmitter phenotype of the same PL neurons, VGAT^FLP::CreER^T::TdTomato^cON/fON mice received a 10-day treatment with either saline or PCP (10 mg/kg/day). Beginning on day 9 of saline/ PCP administration, mice received an injection of tamoxifen each day at 2 pm for 7 consecutive days. Mice were then left untreated for 7 days to enable tamoxifen washout before exposing them to additional

treatments[68]. After the end of the washout period, mice received PCP, METH, or saline for 10 additional days and were sacrificed 2 days after the last injection.

PSEM^308 hydrochloride (Fisher Scientific, 64-252-5) was dissolved in DMSO to obtain a stock solution of 5 mg/ml. Before use, the stock solution was further diluted in saline to a final concentration of 1 mg/ ml. To investigate the effect of acute chemogenetic manipulation on neuronal activity, mice were injected i.p. with PSEM^308 (5 mg/kg) in DMSO/saline (1:4) or DMSO/saline alone (vehicle) 10 min prior to injecting METH, PCP or saline, and were sacrificed 2 h after drug-injection. To investigate the effect of repeated chemogenetic manipulation mice were injected i.p. twice a day for 10 days with either PSEM^308 (5 mg/kg at 10-11 am and 2.5 mg/kg at either 1-2 pm or 4-5 pm) or vehicle. For experiments in which PSEM^308 was administered together with PCP or METH, the first PSEM^308 injection was administered 10 min prior to injecting METH, PCP or saline, and the second PSEM^308 injection occurred 3 h later. For experiments in which the pharmacogenetic treatment occurred after the end of drug treatment, the two PSEM^308 daily injections were administered 6 h apart.

Clozapine (Sigma-Aldrich, C6305) was dissolved in DMSO to obtain a stock solution of 10 mg/ml. The stock solution was further diluted in sterile saline before use to a final concentration of 0.5 mg/ml. DMSO dissolved 1:19 in saline (vehicle) was used as a control solution. Three days after the end of PCP treatment, mice began receiving daily i.p. injections of clozapine (5 mg/kg/day) or DMSO/saline (1:19) alone (vehicle) for 2 additional weeks[14].

## Immunohistochemistry

For immunostaining, coronal sections, 30 μm in thickness, were obtained using a freezing microtome (Leica SM2010R) and stored at −20 °C in a cryoprotectant solution (30% glycerol, 30% ethylene glycol, 20% 0.2 M phosphate buffer). Sections were washed (three times, 15 min each) to remove residues of cryoprotectant solution and permeabilized in 0.3% Triton X-100 in PBS. After a 2 h-incubation period in a blocking solution (5% normal horse serum, 0.3% Triton X-100 in PBS) at 22 °C, sections were incubated o/n on a rotator at 4 °C with primary antibodies diluted in the blocking solution. After washing in 0.3% Triton X-100 in PBS (three times, 15 min each), sections were incubated for 2 h on a rotator at 22 °C with secondary antibodies diluted in blocking solution. After additional washings (three times, 15 min each) in 0.3% Triton X-100 in PBS, sections were mounted with Fluoromount-G (Southern Biotech) containing DRAQ-5 (Thermo Fisher, 62251, 1:1000 dilution) when nuclear staining was needed.

Primary antibodies used in this study were:
rabbit-anti-GABA (Sigma-Aldrich, A2052, RRID:AB_477652, 1:1000), guinea pig-anti-GABA (Sigma-Aldrich, AB175, RRID:AB_91011, 1:500), mouse-anti-GAD67 (Millipore, MAB5406, RRID:AB_227872, 1:500), goat-anti-doublecortin (Santa Cruz, sc-8066, RRID:AB_2088494, 1:300), rabbit-anti-Ki67 (Cell Signaling, 9129, RRID:AB_2687446, 1:300), rabbit-anti-RFP (Avantor VWR, RL600-401-379, RRID:AB_2209751, 1:1000), mouse-anti-cFos (Abcam, ab208942, RRID:AB_2747772, 1:500), rabbit-anti-cFos (Abcam, ab214672, 1:1000), rabbit-anti-PV (Swant, PV27, RRID:AB_2631173, 1:2000), mouse-anti-PV (Millipore, P3088, RRID:AB_477329, 1:1000), rabbit-anti-GFP (Thermo Fisher, A11122, RRID:AB_221569, 1:1000), chicken anti-GFP (Abcam, ab13970, RRID:AB_300798, 1:1000), chicken-anti-mCherry (Abcam, ab205402, RRID:AB_2722769, 1:2000), rabbit-anti-fluorescent gold (Sigma-Aldrich, RRID:AB_2632408, AB153-I, 1:500), mouse-anti-TH (Millipore, MAB318, RRID:AB_2201528, 1:1000), and rabbit-anti-TH (Millipore, AB152, RRID:AB_390204, 1:2000).

Secondary antibodies for immunofluorescence were used at a concentration of 1:500. The following antibodies were from Jackson Immuno Research: Alexa Fluor-488 donkey-anti-rabbit (705-545-003, RRID:AB_2340428), Alexa Fluor-647 donkey-anti-rabbit (711-605-152, RRID:AB_2492288), Alexa Fluor-488 donkey-anti-mouse (715-545-150,

RRID:AB_2340846), Alexa Fluor-488 donkey-anti-goat (705-545-147, RRID:AB_2336933), Alexa Fluor-594 donkey-anti-mouse (715-585-150, RRID:AB_2340854), Alexa Fluor-647 donkey-anti-mouse (715-605-150, RRID:AB_2340862), Alexa Fluor-488 donkey-anti-chicken (703-545-155, RRID:AB_2340375), Alexa Fluor-594 donkey anti-chicken (703-585-155, RRID:AB_2340377), Alexa Fluor-488 donkey anti-guinea pig (706-545-148, RRID:AB_2340472), Alexa Fluor-594 donkey anti-guinea pig (706-586-148, RRID:AB_2340475). The following secondary antibodies were from Life Technologies: Alexa Fluor-555 goat-anti-mouse (A21422, RRID:AB_2535844), Alexa Fluor-555 goat-anti-rabbit (A21429, RRID:AB_2535850).

## TUNEL assay

TUNEL assay was used to detect in situ apoptosis. The assay was performed using the In Situ Cell Death Detection (TUNEL) Kit with TMR Red (Roche, 12156792910) as previously described[12]. Briefly, 30 μm thick PL frozen sections were post-fixed with 1% PFA for 20 min at 22–24 °C and rinsed with PBS (three times, 5 min each). Sections were then permeabilized in 0.1% sodium citrate and 1% Triton X-100 for 1 h at 22–24 °C. After rinsing in PBS (three times, 5 min each), sections were incubated with TUNEL reaction solution according to the vendor's instructions. Incubation was performed in a humidified chamber for 3 h at 37 °C in the dark. Sections were rinsed and mounted with Fluoromount containing DRAQ-5 (1:1000). As positive controls, sections were treated with DNase I (10 U/mL, New England Biolabs, M0303S) for 1 h at 37 °C, rinsed in PBS (three times, 5 min each), and incubated with the TUNEL mixture.

## Fluorescent in situ hybridization (FISH)

FISH was performed using the RNAscope Multiplex Fluorescent v2 Kit (Advanced Cell Diagnostics, 323100) according to the manufacturer's instructions, with a few adjustments. Briefly, the 30-μm fixed brain sections were mounted on Superfrost Plus slides and air-dried in a 50 °C oven for 30 min. Sections were rehydrated in PBS for 2 min and incubated for 5 min in 1X target retrieval solution at 95 °C. After one rinse in distilled water (2 min), sections were dehydrated with 100% ethanol for 5 seconds, air-dried, and incubated for 10 min in 5% hydrogen peroxide at RT. Sections were then incubated in a HybEZ humidified oven at 40 °C with protease III for 30 min, and later with the probe solution for 2 h. After incubation with the probes, slides were incubated o/n in a solution of SSC5X at 22 °C. The following day, sections were incubated with the following solutions in a HybEZ humidified oven at 40 °C with three rinsing steps in between each: amplification Amp-1, 30 min; Amp-2, 30 min; Amp-3, 15 min. For each probe used, sections were incubated in a HybEZ humidified oven at 40 °C with the following solutions: HRP-C1, -C2, or -C3 (depending on the probe) for 15 min, Opal dye of choice for 30 min, and HRP-blocker for 15 min. Opal 520 (Akoya Biosciences, FP1487001KT), Opal 570 (Akoya Biosciences, FP1488001KT) and/or Opal 690 (Akoya Biosciences, FP1497001KT) dyes were used for fluorescent labeling.

Depending on experimental needs, we obtained different levels of signal amplification by changing the dilution of the Opal dyes. To achieve a fully amplified signal, the Opal dyes were used at a dilution of 1:1500. This dilution was used to identify the cell-boundaries of neurons co-expressing mCherry and GAD1 (used to detect mCherry and GAD1 transcripts (Figs. 1e−h and 2e, f, Supplementary Fig. 4) and to quantify the number of VGLUT1+/ GAD1+ PL neurons (Figs. 1k, 3, 4c, d, g, h, 5c, d, and 7c, d, Supplementary Figs. 3, 12k,l, 13d). Because VGLUT1 expression is decreased but not completely lost in neurons that have changed their transmitter identity, obtaining a fully amplified VGLUT1 signal allows detection of these cells.

To obtain unsaturated, puncta-like staining, Opal dyes were diluted 1:12000. This dilution was used to quantify changes in the expression levels of VGLUT1 and VGAT (Figs. 1e−h, 2e, f, Supplementary Fig. 4).

The following probes were used: mouse Probe-Mm-Slc17a7 (VGLUT1) (Advanced Cell Diagnostics, 416631), Probe-mCherry (Advanced Cell Diagnostics, 431201), Probe-Mm-Slc32a1 (VGAT) (Advanced Cell Diagnostics, 319191), Probe-Mm-GAD1 (Advanced Cell Diagnostics, 400951).

For the experiments illustrated in (Fig. 3, Supplementary Figs. 3, 10d, e) RNAscope was followed by standard immuno-fluorescent staining using rabbit-anti-RFP (Avantor VWR, RL600-401-379, 1:1000) or chicken-anti-mCherry (Abcam, ab205402, RRID:AB_2722769, 1:2000) as primary antibodies, and Alexa Fluor-647 donkey-anti-rabbit (711-605-152) or as secondary antibodies.

## Imaging

Images were acquired with a Leica SP5 confocal microscope with a 25×/0.95 water-immersion objective and a z resolution of 1 μm, or with Leica Stellaris 5 with a 20×/0.75 CS2 dry objective and a z resolution of 1 μm for immunohistochemistry and 0.7 μm for RNAscope. Low magnification images were acquired with Leica Stellaris 5 with a 10×/0.40 CS2 dry objective with a z resolution of 2.5 μm.

## Cell counting

All counts were performed by investigators double-blinded to the origin of each image. Either Image-J/Fiji or Imaris9 (Figs. 2c and 6c, Supplementary Figs. 8, 14, 15d–i, 16g, h, 18c, d) was used for cell counting. When using Image-J/Fiji, cell counts were performed by examining all sections within the confocal stacks without maximal projection. When using Imaris9, cell counts were performed semi-automatically using the Spot detection function and the Colocalize Spot plug-in, and later corrected manually. Independently of the software used for analysis, only neurons showing colocalization in at least 3 consecutive z-planes were included in the co-expression group.

PL sections were analyzed from Bregma +2.8 mm to Bregma +1.54 mm, according to the Paxinos Mouse Brain Atlas. PL boundaries were determined based on PL cytoarchitecture, as previously described[69]. Specifically, we used VGLUT1, mCherry or DRAQ5 staining and cellular and laminar morphology as references to identify the boundaries between the PL and its three adjacent regions. To determine the boundaries between the PL and the cingulate cortex, we examined both the arrangement of cells in layers 5 and 6 (which are columnar in the cingulate cortex and either disorderly or horizontal in the PL) and the thickness of layer 2 (which is broader in the PL compared to the cingulate cortex). Similarly, the boundaries between the PL and the medial orbital cortex were identified by examining layer 2 (which shows cells more scattered and more densely concentrated toward layer 1 in the PL than in the medial orbital cortex). The boundaries between the PL and the infralimbic cortex were determined by examining layers 2 to 5 (which are more clearly distinguishable in the PL than in the infralimbic cortex, with layer 2 being broader in the infralimbic cortex than in the PL).

Pilot experiments determined that counting 1 in 6 sections was sufficient to estimate the number glutamatergic and GABAergic cells in the PL. Consequently, to determine the number of PL GABA+/mCherry+ or GAD67+/mCherry+ neurons (Figs. 1c, 2c, and 6c), TdTomato+/PV+ neurons (Supplementary Fig. 5), as well as GAD1+/VGLUT1+ neurons (Fig. 5d), 7-to-8 sections were counted for each mouse brain. The total number of co-expressing neurons was calculated by multiplying the number of counted cells by 6. We later determined that by counting 1 in 12 sections and multiplying the number of counted cells by 12, instead of 6, the final result was not different from that obtained by counting 1 section every 6. We therefore adopted this strategy for the rest of the counts (Figs. 1k, 3d, f, 4d, h and 7c).

To check for equal sampling across experimental groups, we counted the total number of PL mCherry+, TdTomato+ or VGLUT1+ neurons and determined that their number was constant across treatment groups (Supplementary Figs. 1d, 5e, 8b, 12l). For TUNEL

assay, as well as quantification of DCX[+], Ki67[+], and c-fos[+] cells, we scored 1 in 9 PL sections, consequently counting 4-to-6 sections for each mouse brain. To quantify c-fos, GFP, and TH expression in the VTA, we collected 1 in 6 sections from Bregma −2.92 mm to Bregma −3.88 mm, according to the Paxinos Mouse Brain Atlas, thus quantifying 4-5 sections per brain.

## Quantification of VGAT and VGLUT1 mRNA expression
To quantify the expression level of VGLUT1 and VGAT mRNA, we used multiplex RNAscope against mCherry, GAD1, and either VGLUT1 or VGAT in sections from VGLUT1[CRE]::mCherry mice. The RNA signals for mCherry and GAD1 were fully amplified to allow clear detection of mCherry[+] and/or GAD1[+] cell boundaries. To facilitate quantification of mRNA expression levels, we obtained unsaturated, puncta-like RNA-signals for VGLUT1 and VGAT.

After staining, 4-to-7 optical sections (1 μm z step) of each physical section were examined and regions of interest (ROIs) were drawn around the boundaries of mCherry[+], GAD1[+] and GAD1[+]/mCherry[+] cells using the optical section in which the cross-sectional area of the cell was the largest.

VGLUT1 or VGAT expression was quantified using Image-J/Fiji as percent of the ROI occupied by VGLUT1 or VGAT RNA fluorescent signal. For each mouse, we analyzed the ROIs of 25 cells co-expressing mCherry and GAD1, 25 cells expressing only mCherry, and 25 cells expressing only GAD1. These cells were found in sections at different levels of the PL rostrocaudal axis and were distributed across all layers of the PL.

## Stereotaxic injections
4-5-week-old mice were deeply anesthetized using 3-4% vaporized Isoflurane and head-fixed on a stereotaxic apparatus (David Kopf Instruments Model 1900) for all stereotaxic surgeries. Anesthesia was maintained throughout the procedure at a level that prevented reflex response to a tail/toe pinch, using a continuous flow of 1-2% vaporized Isoflurane. Eye drops (Puralube Vet Ointment, Fisher Scientific, 2024927) were placed in each eye to prevent them from drying out, and vitals were checked every 10 min. An incision was made to expose the bregma and lambda point of the skull. A 1 mm drill was used to perforate the skull at the desired coordinates. Stereotaxic coordinates for the injection sites were determined using the Paxinos Brain atlas and adjusted experimentally. Using a syringe pump (PHD Ultra™, Harvard apparatus, no. 70-3007) installed with a microliter syringe (Hampton, 1482452A) and capillary glass pipettes with filament (Warner Instruments, G150TF-4), we infused the brain with 500 nl (for injections in the PL) or 1 μl (for injections in the VTA) of AAV solution for each injection site at a rate of 100 nl/min. To guarantee sufficient AAV expression across the anterior-posterior (AP) extent of the PL the AVV solutions were injected bilaterally at 2 injection sites for each brain hemisphere (from bregma: anterior–posterior (AP), +2.65 mm and +2.25 mm from bregma; mediolateral (ML), ±0.5 mm; dorsal–ventral (DV), −0.8 mm and -1.1 mm from the dura).

After injection of the PL, the pipette was left in place for 8 min to allow diffusion of the virus. When the surgery was completed, the scalp was disinfected with betadine and sutured with tissue adhesive glue (Vetbond tissue adhesive, 1469SB). For post-op pain treatment, mice received an injection of Buprenorphine SR (0.5 mg/kg) or Ethiqa XR (3.25 mg/kg).

When targeting the VTA, we injected the AAV solutions bilaterally (from bregma: AP, −3.2 mm; ML, ±0.5 mm; DV, −4.0 mm from the dura). After injection of the VTA, the pipette was left in place for 16 min to allow diffusion of the virus. Mice that later received optogenetic VTA stimulation were implanted during the same surgical procedure with a fiber optic cannula with Ceramic Ferrule (RWD Life Sciences, R-FOC-F200C-39NA). We used the same coordinates used for AAV

injection with the following modifications: the fiber was implanted at a 10° angle, and the DV coordinate was reduced to −3.9. For selective stimulation of VTA dopaminergic neurons projecting to the PL, AAV were injected into the VTA and fiber optic cannulas were bilaterally implanted over the PL at a 10° angle (from bregma: AP, 2.5 mm; ML, ±0.9 mm; DV, −0.7 mm from the dura). To secure the implants to the skull, the skull was covered with a layer of OptiBond XTR Primer (OptiBond XTR Bottle Primer - 5 ml Bottle. Self-Etching) followed by OptiBond XTR Bottle Universal Adhesive (OptiBond XTR Bottle Universal Adhesive 5 ml Bottle. Self-Etching, Light-Cure). Finally, a thick layer (up to 0.5 cm thick) of Nano-optimized Flowable Composite (Tetric EvoFlow A2 Syringe - Nano-optimized Flowable Composite 1–2 Gram) was used to create a scaffold and secure the optic fiber to the skull. Polymerization of OptiBond XTR Primer, Adhesive, and Flowable Composite was achieved with dental LED light (Fencia Premium Silver LED Light, 5 W).

## Viral constructs
To suppress GAD67 expression in PL glutamatergic neurons we used AAV9-CAG-DIO-shRNAmir-scramble-GFP (6.30E + 13 particles/ml), and AAV9-CAG-DIO-shRNAmir-mGAD1-GFP as control (2.41E + 14 viral particles/ml). pAAV-CAG-DIO-shRNAmir-mGAD1-EGFP and pAAV-CAG-DIO-shRNAmir-Scramble-EGFP plasmid[12] were produced by Vector Biolabs and AAV9 vectors were packaged in the Salk Institute Viral Vector Core. The shRNA sequence for mouse GAD1 is 5′-GTCTACAGTCAACCAGG ATCTGGTTTTGGCCACTGACTGACCAGATCCTTTGACTGTAGA-3′. Validation of AAV-CAG-DIO-shRNAmir-scramble-GFP and AAV-CAG-DIO-shRNAmir-mGAD1-GFP can be found in ref. 12.

To activate PL PV[+] neurons we used AAV9-syn-FLEX-rev-PSAML141F,Y115F:5HT3HC-IRES-GFP (4.61E + 12 viral particles/ml). rAAV-syn::FLEX-rev::PSAML141F,Y115F:5HT3HC-IRES-GFP was a gift from Scott Sternson (Addgene plasmid # 32477; http://n2t.net/addgene:32477; RRID:Addgene_32477)[70], and the AAV9 vector was packaged in the Salk Viral Vector Core.

To inhibit dopaminergic neurons in the VTA, we used AAV9-syn::FLEX-rev::PSAML141F,Y115F:GlyR-IRES-GFP (2.1E + 12 viral particles/ml). rAAV-syn::FLEX-rev::PSAML141F,Y115F:GlyR-IRES-GFP was a gift from Scott Sternson (Addgene plasmid # 32481; http://n2t.net/addgene:32481; RRID:Addgene_32481)[70].

To optogenetically stimulate VTA dopaminergic neurons, we used AAV5-EF1a-double floxed-hChR2(H134R)-EYFP-WPRE-HGHpA (1E + 12 viral particles/ml) and AAV5-Ef1a-DIO EYFP as control (2.3E + 12 viral particles/ml). pAAV-EF1a-double floxed-hChR2(H134R)-EYFP-WPRE-HGHpA and pAAV-Ef1a-DIO EYFP were a gift from Karl Deisseroth (Addgene viral prep # 20298-AAV5; http://n2t.net/addgene:20298; RRID:Addgene_20298; and Addgene viral prep # 27056-AAV5; http://n2t.net/addgene:27056; RRID:Addgene_27056).

## Retrograde tracing
For retrograde tracing, 50 nl of Fluoro-gold (Hydroxystilbamidine Fluoro-Gold™), 4% in $H_2O$, (Biotium, 80023) were stereotaxically injected at 50 nl/min into the ventral nucleus accumbens (from bregma: AP, +1.40 mm; ML, ±1.20 mm; and DV, −4.20 mm from the dura). The glass pipette was left in place for 10 min after injection.

## Optogenetic stimulation
The first stimulation session occurred 4-5 weeks after surgery. After being moved to the room where stimulation was performed, mice were acclimatized to the room for 1 h. Stimulation occurred in the home cage while the mouse was allowed to move. The ceramic ferrule protruding from the animal's head was coupled to a DPSS blue light laser (473 nm Blue DPSS Laser with Fiber Coupled, BL473T3-050FC, with ADR-700A Power Supply; Shanghai Laser & Optics Century) via custom-made patch cords. Patch cords were assembled by epoxying

optical fibers (200 μm, 0.39 numerical aperture, Thorlabs, FT200EMT) to Fiber Connectors FC/PC with Ceramic Ferrule (Thorlabs, 30140E1) and polishing the optic fiber with a fiber polishing kit (Thorlabs) to achieve a minimum of 85% transmission. The laser power was measured before each mouse/stimulation session using a Compact Power and Energy Meter Console, Digital 4" LCD (Thorlabs, PM 100D). Before each mouse/stimulation session, we attached an unused fiber optic cannula to the optic cable and measured the laser power at the tip of the fiber optic cannula using the Compact Power and Energy Meter Console, Digital 4" LCD and adjusted the laser power to $8.5 \pm 1$ mW at the tip of the fiber optic cannula. Each session of optogenetic stimulation lasted 1 h, during which 80 sets of laser stimulation, each in turn consisting of 30 bursts at 4 Hz of 5 pulses of 4 ms at 20 Hz, were delivered (modified from[32]). For c-fos quantification experiments, mice received a single session of optogenetic stimulation and were sacrificed 1 h after the end of the session. For the 10-days stimulation protocol, mice received a daily session of 1 h stimulation between 11 am and 5 pm and were sacrificed 2 days after the last stimulation.

### Patch clamp slice electrophysiology
To confirm the efficacy of ChR2 stimulation in VTA dopaminergic neurons we performed whole-cell patch clamp recordings on brain slices obtained from the VTA of DAT$^{CRE}$ mice injected with AAV5-EF1a-double floxed-hChR2(H134R)-EYFP-WPRE-HGHpA. 3 weeks after the injection of the AAV, mice were euthanized using isoflurane inhalation, and coronal brain slices (250 μm) containing the VTA were prepared using a vibratome (Leica VT1200S), following standard procedures[71]. VTA cells expressing YFP were visualized on a differential interference microscope with a 4X or a 40X objective (Olympus BX61WI using epifluorescence), and were selectively patched. Recordings were performed in the whole cell configuration using borosilicate glass pipettes (3-4 MΩ) filled with a solution containing: 125 mM Potassium gluconate, 4 mM NaCl, 10 mM HEPES, 0.5 mM EGTA, 20 mM KCl, 4 mM Mg-ATP, 0.3 mM Na$_3$-GTP, 10 mM Na-phosphocreatine. VTA cells were optogenetically activated using an LED light source (Excelitas XT720L), delivering blue light (473 nm, 4 ms pulses, ~0.5 mW) via a 40X objective. Liquid junction potentials were not compensated. Signals were amplified with an Axopatch 700B amplifier, sampled at 10 kHz using a Digidata 1550, and recorded with Clampex 10.4 (Molecular Devices). Data Analyses were performed offline using Clampfit 11.2 (Molecular Devices). Only cells with stable access resistance <30 MΩ throughout the recording period were included in the analysis.

### Drug-induced locomotor sensitization
Locomotor activity was measured as total distance traveled in the home cage during the 90 min immediately after a single injection of PCP, METH or saline. Mouse movements were recorded using a camera suspended 2 m above the home cage and were automatically scored using AnyMaze 5.2 (Stoelting, Wood Dale, IL, USA). In experiments aimed at preventing drug-induced changes in neurotransmitter phenotype by combining PCP, METH, or saline treatment with chemogenetic activation of PV+ interneurons, or by suppressing the gain of GABA with shGAD1 interference, locomotor activity was measured on the first day of treatment (i.e. immediately after the first drug/saline injection, DAY1) and on the last day of treatment (i.e. immediately after the tenth drug/saline injection, DAY10).

For experiments in which mice received either clozapine treatment or chemogenetic manipulation of PL activity after the end of PCP-, METH- or saline-treatment, locomotor quantification was performed at the end of the experimental timeline, two days after the spontaneous alternation task (SAT). In this case, baseline locomotor activity was first recorded in the home cage for 90 min, after which all mice received an acute injection of PCP 10 mg/kg (*PCP challenge*) or METH 1 mg/kg (*METH challenge*).

### Novel object recognition test (NORT)
The novel object recognition test was used to assess recognition memory performance and performed as described[72] with the modifications outlined below. The test was conducted in an open field box made of dark gray plastic 40 × 40 × 21 cm, dimly illuminated (30–40 lux). In the 3 days before the test, mice were acclimatized to the open field box for 5 min/day. 24 h after the last acclimatization session, mice were placed back in the open field with two identical objects and allowed to explore the 2 objects for 10 minutes (familiarization phase). 24 h after the familiarization phase, memory retention was tested by placing the mouse back in the open field where one of the familiar objects had been replaced by a novel object (test phase). The mouse was left to explore the 2 objects for 12 min and video recordings were collected with a camera (Sony HDR-CX405) suspended 1 m above the apparatus. Because mice (including saline-treated controls) often failed to reach an accepted exploration criterion[72] (20 s of total exploration in the 10 min of the test), we extended the time available for exploration from 10 to 12 min, and mice that did not reach the criterion in less than 12 min were excluded. Mice were also excluded if they moved or overturned one of the two objects. The time spent exploring each of the two objects (novel and familiar) was manually scored using BORIS software (version v. 5.0.1[73]) by investigators blinded to the mouse's previous treatment-history. The mouse was considered to be exploring the object when it had its nose directed toward the object at a distance of <1 cm. Chewing or climbing on the object was not considered an exploratory behavior. Exploratory behavior was scored until the mouse reached the criterion of 20 s spent exploring the two objects. We calculated a recognition index (RI) as the percent of time spent exploring the novel object relative to the total time spent exploring both objects (novel and familiar) [RI=time exploring novel object/(time exploring novel object + time exploring familiar object)]. Total exploration time and time to reach the criterion were also recorded.

### Spontaneous alternation task (SAT)
The spontaneous alternation task was performed as described[19,74] and used to measure immediate working memory performance. The test was conducted in a T-shaped, dark gray plastic maze, composed of 3 arms 30 cm long, 9 cm wide and 20 cm tall. Mice naïve to the maze were placed at the end of one of the 3 arms, and left free to explore the maze for 8 min. Exploration was recorded with a camera suspended 1 m above the apparatus, and the series of arm entries was scored using BORIS software by investigators blinded to the mouse's previous treatment history. The mouse was considered to have entered an arm of the maze only when both the forelimbs and hindlimbs were completely within the arm. Alternation was defined as successive entries into the three arms of the maze on overlapping triplet sets. Alternation percent was calculated as ratio of actual alternation to total possible alternations (defined as the total number of arm entries minus two), multiplied by 100. Total arm entries were also scored to compare the degree of exploration across treatment groups.

### Statistics
Statistical analyses of the data were performed using Prism 9 software. The Excel Real Statistics package was used for non-parametric Aligned Rank Transform (ART) ANOVA. The Shapiro–Wilk test was used to assess whether the data were normally distributed. All statistical tests were two-tailed. Details about the number of biological replicates and statistical tests used for each experiment are reported in the figure legends and in the Supplementary Fig. legends. Measurements were always taken from biological replicates, except for measurements of locomotor activity on first and last day of treatment, which were on the same mice. Means and SEMs are reported for all experiments.

**Reporting summary**

Further information on research design is available in the Nature Portfolio Reporting Summary linked to this article.

## Data availability

The data generated in this study (including details about the statistical analysis and tabulated single data points) are provided in the Supplementary Information/Source Data files. Source data are provided with this paper.

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

## Acknowledgements

We thank Dr. Cory Root, Dr. Davide Dulcis, and Dr. Thomas Hnasko for providing DAT-IRES-Cre mice (JAX#006660). We are grateful to Dr. Cory Root, Donghyung Lee and James Howe VI for providing the materials and guidance for the optogenetic experiments. We thank Dr. Scott Sternson for donating the rAAV-syn::FLEX-rev::PSAML141F,Y115F:5HT3HC-IRES-GFP and the rAAV-syn::FLEX-rev:: PSAML141F,Y115F:GlyR-IRES-GFP plasmid to Addgene. We thank Dr. Karl Deisseroth for donating the pAAV-EF1a-double floxed-hChR2(H134R)-EYFP-WPRE-HGHpA and pAAV-Ef1a-DIO EYFP plasmids to Addgene. We thank Alexander Glavis-Bloom for assisting with genotyping and for technical support; Hannah Kim, Ramiz Ahmed, and Tianna Huang for assisting with experimental procedures. We thank. Cory Root, Davide Dulcis, and Samuel Barnes for their thoughtful reviews and comments on an earlier version of this manuscript. This work was supported by a R21 (CEBRA) grant from NIDA DA048633 and by R21 NIDA DA050821 (to N.C.S.) and by the Overland Foundation (to N.C.S.).

## Author contributions

M.P. and N.C.S. conceived the study, designed the experiments, interpreted the results and wrote the manuscript. M.P., A.H., A.T. and H.J. performed experiments and analyses. M.P. designed and performed the chemogenetic and ontogenetic experiments. H-q.L and S.G. contributed to experimental design and data interpretation. B.K.L. designed the electrophysiological experiments and provided valuable input on the manuscript.

## Competing interests

The authors declare no competing interests.
