## [Peer Review File · Nature Communications]

Drug-induced change in transmitter identity is a shared mechanism generating cognitive deficitsREVIEWER COMMENTS

Reviewer #1 (Remarks to the Author):

The manuscript by Pratelli and colleagues describe a common mechanism by which drugs of abuse with different pharmacological mechanisms such as methamphetamine and PCP cause glutamatergic neurons in the prelimbic PFC gain a GABAergic phenotype.

The paper is well written and the experiments are elegant, well designed and with appropriate controls.

The findings are very interesting and of relevance to the fields of addiction and neuropsychiatric disorders. And as any exciting finding the results generate many questions, some outside the scope of this manuscript but others that are germane to the present research.

The main concerns of the reviewer are described below:

1. The authors shows that activity of glutamate neurons projecting to VTA is increased by PCP and meth, and the expression of Cfos is also increased. It is the activity of this immediate early gene the main molecular mechanisms by which the neurons gain the GABAergic phenotype? Drugs of abuse increase Cfos expression in all glutamatergic neurons in the PFC, so if is only the juxtaposition of Cfos expression AND VTA-PFC hyperactivity what underlies the gain of GABAergic phenotype in some glutamatergic neurons, the manuscript will benefit from the authors stating this clearly.
2. Even when the word mechanisms is included in the title, the manuscript doesn't describe the cellular/molecular mechanisms underlying the gain in GABAergic phenotype. What genes are changed by the combination of CFos and VTA hyperactivity? Including putative mechanisms in the discussion will improve the impact of the manuscript.
3. The authors describe that hyperactivity of VTA neurons contribute to the expression of the GABAergic phenotype on glutamate expressing neurons. Does only VTA hyperexcitability induces this new phenotype expression or hyperexcitability of any other input to these glutamatergic expressing neurons will do it? Experiments demonstrating that either hyperexcitability of other inputs to these glutamatergic neuron (i.e., cortical inputs, thalamic inputs, etc) do or do not induce GABAergic phenotype expression will clarify a little bit more thus phenomenon. .
4. The author's document very well the expression of the GABAergic phenotype in the glutamatergic neurons of the prelimbic PFC, however there is not demonstration that these neurons release GABA. It is possible that the expression of the GABA phenotype doesn't have consequences at cellular or synaptic level? The authors will need to demonstrate that this new GABA phenotype in the glutamatergic neurons is been translated into real electrophysiological effects.
5. It is not clear which type of VTA projection neurons to the prelimbic PFC glutamatergic neurons are involved in the hyperactivity and resulting expression of GABAergic phenotype. Are they only DA expression neurons, DA/Glutamate expression neurons?, A mix?
6. The author's mention that exposure to PCP or meth decreases expression of PV however other research has shown that meth treatment increases GABAergic activity (Campanac et al., 2013; Morshedi et al., 2007) and decrease in PV expression may not necessary translate in decrease in GABA release. Please discuss.

Reviewer #2 (Remarks to the Author):

Pratelli and colleagues describe an interesting set of experiments examining neurotransmitter switching in medial prefrontal cortex in response to two drugs of abuse – phencyclidine (PCP) and methamphetamine (METH). The authors show that both PCP and METH induce the synthesis and expression of GABA in prelimbic glutamatergic neurons. This increase was persistent and causally linked with cognitive (memory) deficits observed after repeated PCP/METH exposure. Using

chemogenetic and optogenetic approaches, the authors show that the increase in GABA production is driven by dopamine signaling originating in VTA, and is reliant on hyperactivity of local glutamatergic neurons in PL. The authors show that treatment with clozapine can reverse PCP/METH-induced upregulation of GABA in PL neurons and associated memory deficits.

Overall, the experiments in this paper are well designed and executed, the analyses appear to be appropriate, and the manuscript is written clearly. The approaches used are state of the art and appropriate for the questions under investigation. Although there is previous evidence (mostly from the same authors) of neurotransmitter switching in response to stress and other environmental stimuli, this paper offers novel insights neural plasticity associated with drugs of abuse. I am very enthusiastic about this paper, but below highlight several concerns that I believe should be addressed.

1. The authors speculate in the Discussion that 'new' GABA production in PL glutamate neurons might influence nucleus accumbens function and associated behavioral outcomes. Although I agree that this is likely, the paper in its current form has little data to support this. Indeed, the only data examining the PL-accumbens projection is presented in Supplementary Fig. 8, where it is reported that there is an increase in GABA expression in glutamatergic PL-NAc neurons following PCP/METH. Although these data are compelling, they are insufficient to support the claim that "gain of GABA by PL neurons projecting to NAc may contribute to the reduction in NAc firing rates and disruption of cortex-accumbens synchronization" (although I recognize that this statement is posed somewhat speculatively). It would be powerful if the authors could demonstrate that increased GABA in accumbens-projecting PL neurons is causally linked with the behaviors of interest; one might speculate that this pathway might be important for the locomotor sensitization behavior, although their role in cognitive outcomes would be of interest too. I recognize that these experiments are not trivial, but given the centrality of the PL-NAc projection to the authors' interpretation of the data in the Discussion, I believe they would be useful. Alternatively, the authors should de-emphasize their discussion of this pathway. Relatedly, given the data demonstrating that these effects are mediated by dopamine, the authors might wish to discuss a relevant paper that showed that PL-NAc neurons are recruited in a dopamine-dependent manner to drive reward behaviors for cocaine (PMID: 27535915).
2. The demonstration that neurotransmitter switching can be blocked by inhibiting VTA DA neurons, and recapitulated by optogenetic stimulation of DA cells is compelling. As the authors note in the Discussion, these data do not speak directly to whether these effects are mediated via direct dopaminergic inputs to PL. A couple of questions: 1) Could the authors please discuss how closely the optogenetic stimulation parameters used here reflect activity patterns produced by either METH or PCP exposure? 2) To show that the effects are indeed dopamine mediated, did the authors consider showing that the effects of stimulating VTA neurons are blocked by a selective dopamine receptor antagonist delivered to PL? 3) Relatedly, why stimulate the VTA cell bodies and not terminals in PL, or use a retro approach to label PL-projecting VTA DA neurons?
3. The persistence of changes in GABA production and cell activity are intriguing, and perhaps speak to the chronic nature of drug addiction. I am particularly curious about the increase in PL neuronal activity (gauged by cfos): where animals perfused at the same time that they previously received drug? i.e. could this increase in activity reflect a time-of-day specific increase in activity of these neurons? Do the authors have evidence indicating that these neurons exhibit increased activity across the entire day/night cycle following drug treatment? Surely there would be other behavioral phenotypes associated with chronic high activity of these neurons? For example, do these rats consume more food?
4. The clozapine experiments are interesting, but the mechanism(s) governing its effects – and its interaction with local PV+ neurons - are not clear to me. Do the authors propose that clozapine somehow increases the activity of PV+ neurons? If so, how? Can the authors also comment on why they think clozapine did not reverse drug-associated hyperlocomotion (Fig 6g) but activation of PV+ neurons did (Fig 7i)?

5. What relevance do these data have to the neural mechanisms governing the rewarding properties of these drugs? Also, some acknowledgement should be made in the paper that different effects might be expected in a model of drug self-administration vs. the experimenter-administered model utilized here.

Reviewer #3 (Remarks to the Author):

This is an interesting study, in which the authors demonstrated phencyclidine (PCP) and methamphetamine (METH), acting on different targets in the brain, induced the same population of glutamatergic neurons in the medial prefrontal cortex (mPFC) to gain GABA and reduce the expression of the vesicular glutamate transporter (VGLUT), which leads to memory deficits. Additional results show that dopaminergic neurons in the ventral tegmental area (VTA) contributed to the gain of GABA in the mPFC. Drug-induced hyperactivity in the mPFC changed the transmitter identity, which can be reversed and repair the memory deficits. Overall, their experimental design is coherent, the data appear to be solid, the conclusions are reasonable, and the manuscript is well written. Below are a few minor points for the authors to consider.

1. The mPFC is a major hub for cognitive control (Line 56), and the study focused on the prelimbic subregion (PL). However, the mPFC consists of several areas including both the prelimbic and infralimbic cortex that are involved in different cognitive performances. An explanation would be helpful for readers to understand the selection of brain regions.
2. In the supplementary Fig.10, the authors performed chemogenetic inhibition of VTA dopaminergic neurons, then used c-fos expression to examine the effect of virus. To detect the change of excitability in VTA dopaminergic neurons, action potential firing is a more direct indicator.
3. In the supplementary Fig.11, to test the effectiveness of YFP-ChR2, it's necessary to verify the optogenetic stimulation in the brain slices using electrophysiological approach.
4. In the supplementary Fig.12, both PCP and METH increased c-fos expression in PL glutamatergic neurons. To confirm that chronic drug treatment increases the excitability of PL glutamatergic neurons, spontaneous and evoked action potential recordings are preferred.
5. PCP mainly interacts with NMDA receptor, and METH mainly interact with dopamine transporter. To help understand the mechanism, a discussion is needed about how PCP and METH induce similar effects on the excitability of VTA dopaminergic neurons and PL glutamatergic neurons, and the roles of VTA-mPFC circuit in regulating cognitive behaviors.

REVIEWER COMMENTS

Reviewer #1 (Remarks to the Author):

The manuscript by Pratelli and colleagues describe a common mechanism by which drugs of abuse with different pharmacological mechanisms such as methamphetamine and PCP cause glutamatergic neurons in the prelimbic PFC gain a GABAergic phenotype.

The paper is well written and the experiments are elegant, well designed and with appropriate controls.

The findings are very interesting and of relevance to the fields of addiction and neuropsychiatric disorders. And as any exciting finding the results generate many questions, some outside the scope of this manuscript but others that are germane to the present research.

The main concerns of the reviewer are described below:

1. The authors shows that activity of glutamate neurons projecting to VTA is increased by PCP and meth, and the expression of Cfos is also increased. It is the activity of this immediate early gene the main molecular mechanisms by which the neurons gain the GABAergic phenotype? Drugs of abuse increase CFos expression in all glutamatergic neurons in the PFC, so if is only the juxtaposition of Cfos expression AND VTA-PFC hyperactivity what underlies the gain of GABAergic phenotype in some glutamatergic neurons, the manuscript will benefit from the authors stating this clearly.

We used c-fos expression as a reporter of increased electrical activity. We found that treatment with PCP or METH leads to increased c-fos expression in a subset of PL glutamatergic neurons (unfortunately, we did not identify the projection targets of neurons that show increased c-fos expression) and that this increase in c-fos expression is linked to gain of the GABAergic phenotype. Since we find that suppressing electrical activity prevents the gain of this phenotype, we conclude that the electrical activity reported by c-fos is necessary for this change in transmitter identity. It remains to be determined whether increased electrical activity by itself is sufficient for the change in transmitter identity or whether, for example, some specific property of the switching neurons is also required. While we are aware that c-fos can be itself involved in some METH-induced changes in gene expression (Cadet, McCoy et al. 2002), whether expression c-fos alone is sufficient to change transmitter identity is unknown. We have clarified this point in the Discussion (lines 403 to 405).

2. Even when the word mechanisms is included in the title, the manuscript doesn't describe the cellular/molecular mechanisms underlying the gain in GABAergic phenotype. What genes are changed by the combination of CFos and VTA hyperactivity? Including putative mechanisms in the discussion will improve the impact of the manuscript.

Our analysis revealed that the combination of PL cFos expression and VTA hyperactivity led to increased expression of GAD1 and VGAT transcripts, associated with increased GAD67 and

GABA, as well as decreased expression of VGLUT1 transcripts (Figure 1 and Figure 2). We have added a paragraph to the Discussion, outlining potential mechanisms driving the change in transmitter identity, based on our previous studies of the molecular mechanisms of activity-dependent changes in transmitter phenotype (lines 412 to 419).

3. The authors describe that hyperactivity of VTA neurons contribute to the expression of the GABAergic phenotype on glutamate expressing neurons. Does only VTA hyperexcitability induces this new phenotype expression or hyperexcitability of any other input to these glutamatergic expressing neurons will do it? Experiments demonstrating that either hyperexcitability of other inputs to these glutamatergic neuron (i.e., cortical inputs, thalamic inputs, etc) do or do not induce GABAergic phenotype expression will clarify a little bit more thus phenomenon.

We thank the reviewer for this fascinating question. We are aware of the interesting evidence suggesting that thalamic and hippocampal inputs to the mPFC cause mPFC hyperactivity in response to PCP (Jodo, Suzuki et al. 2005, Kargieman, Santana et al. 2007, Santana, Troyano-Rodriguez et al. 2011). There are many inputs to the glutamatergic neurons in the prelimbic cortex (Ährlund-Richter, Xuan et al. 2019). Because we have found that VTA-to-PL projections are not sufficient to produce the gain of the GABAergic phenotype (see response to point 5, below), other inputs to the PL are required for this change in transmitter identity to occur. Working out the details of the circuits generating activity that contributes to the GABAergic phenotype is now the subject of another project. We have clarified this point in the text (lines 383 to 387).

4. The author's document very well the expression of the GABAergic phenotype in the glutamatergic neurons of the prelimbic PFC, however there is not demonstration that these neurons release GABA. It is possible that the expression of the GABA phenotype doesn't have consequences at cellular or synaptic level? The authors will need to demonstrate that this new GABA phenotype in the glutamatergic neurons is been translated into real electrophysiological effects.

Stimulating neurons that have gained the GABAergic molecular phenotype (as distinct from neurons that normally co-express GABA and glutamate) while recording from their postsynaptic partners appears straightforward but is actually quite challenging. There is no switching-neuron-specific molecular marker for presynaptic neuron identification, nor a molecular marker for identification of the specific postsynaptic partner population. We think this set of experiments constitutes an entire project in itself. However, our finding that the molecular phenotype is correlated with cognitive deficits, and that overriding the molecular phenotype prevents the cognitive deficits, is consistent with a change in the electrophysiological signature of the synapses.

5. It is not clear which type of VTA projection neurons to the prelimbic PFC glutamatergic neurons are involved in the hyperactivity and resulting expression of GABAergic phenotype. Are they only DA expression neurons, DA/Glutamate expression neurons? A mix?

We have now done experiments to test whether VTA-to-PL projections are directly responsible (via monosynaptic inputs) for the gain of the GABAergic phenotype, so that we could determine the transmitter phenotype of these neurons. However, we found that stimulation of VTA projections to the PL is not sufficient to induce the change in neurotransmitter phenotype, indicating that indirect, multi-synaptic pathways are required. These results are presented in a new supplementary figure (Supplementary Fig. 13) and in the text (lines 251 to 262). Therefore, while it's known that both TH/only and TH/VGLUT2 neurons from the VTA project to the PL (Yamaguchi, Wang et al. 2011), based on our results, none of these neuronal types is alone responsible for the gain of GABAergic phenotype in PL glutamatergic neurons.

6. The author's mention that exposure to PCP or meth decreases expression of PV however other research has shown that meth treatment increases GABAergic activity (Campanac et al., 2013; Morshedi et al., 2007) and decrease in PV expression may not necessary translate in decrease in GABA release. Please discuss.

The comment appears to refer to a hypothesis we advanced in the Discussion (lines 407-409), suggesting that during withdrawal, drug-induced impairment of the proper function of PV neurons in the PL may contribute to maintain a status of glutamatergic hyperactivity. This hyperactivity, could contribute to retention of the newly acquired GABAergic phenotype. We have removed this suggestion in light of new evidence (Supplementary Fig. 17c) showing that PL hyperactivity after the end of drug-treatment shows time-of-day dependent variation. Thus, unless drug-induced impairment of PL PV neuron function is also time-of-day dependent, its contribution to retention of the newly acquired GABAergic phenotype is less probable.

Morshedi et al., 2007 (Morshedi and Meredith 2007) reported that sensitization to amphetamine reduces the number of IHC-detectable PV+ neurons in layer V of the rat prelimbic cortex, which would support our initial hypothesis. They also quantified the number of neurons positive for both PV and c-fos, and found no difference between saline and amphetamine-treated mice. The fraction of total PV+ neurons that was c-fos+ was instead increased in amphetamine-treated mice presumably because of the decrease in the total number of PV+ neurons. As Morshedi and co-workers quantified c-fos after an amphetamine challenge, the result could not be considered fully representative of the baseline state of prelimbic PV neurons after amphetamine sensitization.

The study performed by Campanac et al., 2013 (Campanac and Hoffman 2013) reported hyperactivity of PV+ prelimbic neurons after long-term withdrawal (10–13 days) from cocaine. The use of a drug other than METH or PCP may explain the different results. Alternatively, the number of neurons expressing PV decreases, but the activity of the remaining PV neurons increases as form of compensation.

REFERENCES:

ÄHRLUND-RICHTER, S.; XUAN, Y.; VAN LUNTEREN, J. A.; KIM, H. *et al.* A whole-brain atlas of monosynaptic input targeting four different cell types in the medial prefrontal cortex of the mouse. **Nat Neurosci**, 22, n. 4, p. 657-668, Apr 2019.

CADET, J. L.; MCCOY, M. T.; LADENHEIM, B. Distinct gene expression signatures in the striata of wild-type and heterozygous c-fos knockout mice following methamphetamine administration: evidence from cDNA array analyses. **Synapse**, 44, n. 4, p. 211-226, Jun 15 2002.

CAMPANAC, E.; HOFFMAN, D. A. Repeated cocaine exposure increases fast-spiking interneuron excitability in the rat medial prefrontal cortex. **J Neurophysiol**, 109, n. 11, p. 2781-2792, Jun 2013.

JODO, E.; SUZUKI, Y.; KATAYAMA, T.; HOSHINO, K. Y. *et al.* Activation of medial prefrontal cortex by phencyclidine is mediated via a hippocampo-prefrontal pathway. **Cereb Cortex**, 15, n. 5, p. 663-669, May 2005.

KARGIEMAN, L.; SANTANA, N.; MENGOD, G.; CELADA, P. *et al.* Antipsychotic drugs reverse the disruption in prefrontal cortex function produced by NMDA receptor blockade with phencyclidine. **Proceedings of the National Academy of Sciences**, 104, n. 37, p. 14843-14848, 2007.

MORSHEDI, M. M.; MEREDITH, G. E. Differential laminar effects of amphetamine on prefrontal parvalbumin interneurons. **Neuroscience**, 149, n. 3, p. 617-624, Nov 9 2007.

SANTANA, N.; TROYANO-RODRIGUEZ, E.; MENGOD, G.; CELADA, P. *et al.* Activation of thalamocortical networks by the N-methyl-D-aspartate receptor antagonist phencyclidine: reversal by clozapine. **Biol Psychiatry**, 69, n. 10, p. 918-927, May 15 2011.

YAMAGUCHI, T.; WANG, H. L.; LI, X.; NG, T. H. *et al.* Mesocorticolimbic glutamatergic pathway. **J Neurosci**, 31, n. 23, p. 8476-8490, Jun 8 2011.

Reviewer #2 (Remarks to the Author):

Pratelli and colleagues describe an interesting set of experiments examining neurotransmitter switching in medial prefrontal cortex in response to two drugs of abuse – phencyclidine (PCP) and methamphetamine (METH). The authors show that both PCP and METH induce the synthesis and expression of GABA in prelimbic glutamatergic neurons. This increase was persistent and causally linked with cognitive (memory) deficits observed after repeated PCP/METH exposure. Using chemogenetic and optogenetic approaches, the authors show that the increase in GABA production is driven by dopamine signaling originating in VTA, and is reliant on hyperactivity of local glutamatergic neurons in PL. The authors show that treatment with clozapine can reverse PCP/METH-induced upregulation of GABA in PL neurons and associated memory deficits.

Overall, the experiments in this paper are well designed and executed, the analyses appear to be appropriate, and the manuscript is written clearly. The approaches used are state of the art and appropriate for the questions under investigation. Although there is previous

evidence (mostly from the same authors) of neurotransmitter switching in response to stress and other environmental stimuli, this paper offers novel insights neural plasticity associated with drugs of abuse. I am very enthusiastic about this paper, but below highlight several concerns that I believe should be addressed.

1. The authors speculate in the Discussion that 'new' GABA production in PL glutamate neurons might influence nucleus accumbens function and associated behavioral outcomes. Although I agree that this is likely, the paper in its current form has little data to support this. Indeed, the only data examining the PL-accumbens projection is presented in Supplementary Fig. 8, where it is reported that there is an increase in GABA expression in glutamatergic PL-NAc neurons following PCP/METH. Although these data are compelling, they are insufficient to support the claim that "gain of GABA by PL neurons projecting to NAc may contribute to the reduction in NAc firing rates and disruption of cortex-accumbens synchronization" (although I recognize that this statement is posed somewhat speculatively). It would be powerful if the authors could demonstrate that increased GABA in accumbens-projecting PL neurons is causally linked with the behaviors of interest; one might speculate that this pathway might be important for the locomotor sensitization behavior, although their role in cognitive outcomes would be of interest too. I recognize that these experiments are not trivial, but given the centrality of the PL-NAc projection to the authors' interpretation of the data in the Discussion, I believe they would be useful. Alternatively, the authors should de-emphasize their discussion of this pathway. Relatedly, given the data demonstrating that these effects are mediated by dopamine, the authors might wish to discuss a relevant paper that showed that PL-NAc neurons are recruited in a dopamine-dependent manner to drive reward behaviors for cocaine (PMID: 27535915).

We thank the reviewer for the suggestion. We did not test whether the increased GABA in PL neurons projecting to the NAc cause the behaviors of interest. We agree that the experiments to test this hypothesis would be interesting, but we think that they are complex. Accordingly, we have de-emphasized the role of the PL-NAc pathway in the discussion by removing lines 366-369. We have added a discussion of the relevance of PL-to-NAc projection for reward and drug-seeking behavior, suggesting that the gain of GABA may influence reinstatement of drug-seeking, but that testing this hypothesis would require self-administration experiments and demonstration that self-administration induces the same change in neurotransmitter phenotype as observed after experimenter-administration of drugs (lines 370 to 375).

2. The demonstration that neurotransmitter switching can be blocked by inhibiting VTA DA neurons, and recapitulated by optogenetic stimulation of DA cells is compelling. As the authors note in the Discussion, these data do not speak directly to whether these effects are mediated via direct dopaminergic inputs to PL. A couple of questions:

1) Could the authors please discuss how closely the optogenetic stimulation parameters used here reflect activity patterns produced by either METH or PCP exposure?

Our protocol of VTA optogenetic mimics the increase in the levels of mesolimbic dopamine induced by the intake of addictive drugs, rather than replicating the effect of PCP and METH on

VTA neuron firing that shows dynamics that change as a function of time. We reproduced, in a non-operant set up, a protocol of optogenetic VTA dopamine self-stimulation (Pascoli, Terrier et al. 2015) that increases dopamine release in the mesolimbic system. This protocol was shown by others to induce self-administration, CPP and addiction-like behaviors. We have clarified this in the text (lines 237 to 238 and lines 377 to 379)

2) To show that the effects are indeed dopamine mediated, did the authors consider showing that the effects of stimulating VTA neurons are blocked by a selective dopamine receptor antagonist delivered to PL?

We appreciate the question. We have now determined that the change in transmitter phenotype of PL glutamatergic neurons observed after optogenetic stimulation of VTA dopaminergic neurons is unlikely to be mediated by dopamine signaling in the PL, as selective stimulation of VTA dopaminergic projections to the PL is not sufficient to induce the gain of GABA. Please see our response to Reviewer 1, question 3 and question 5.

3) Relatedly, why stimulate the VTA cell bodies and not terminals in PL, or use a retro approach to label PL-projecting VTA DA neurons?

We thank the reviewer for this question and appreciate the relevance of the proposed experiment. We initially stimulated VTA dopaminergic cell bodies to determine whether a hyperdopaminergic state mimicking that induced by drug-intake was sufficient for glutamatergic neurons in the PL to gain GABA. We have now selectively stimulated dopaminergic fibers from VTA neurons projecting to the PL, and found that this is not sufficient to induce the change in transmitter phenotype (Supplementary Fig. 13 and text lines 251 to 262). Please see also our response to Reviewer 1, question 3 and question 5.

3. The persistence of changes in GABA production and cell activity are intriguing, and perhaps speak to the chronic nature of drug addiction. I am particularly curious about the increase in PL neuronal activity (gauged by cfos): were animals perfused at the same time that they previously received drug? i.e. could this increase in activity reflect a time-of-day specific increase in activity of these neurons? Do the authors have evidence indicating that these neurons exhibit increased activity across the entire day/night cycle following drug treatment? Surely there would be other behavioral phenotypes associated with chronic high activity of these neurons? For example, do these rats consume more food?

We appreciate the reviewer's insights and questions. Animals were perfused at the same of day that they previously received the drug, within a 5-h window. We have now examined c-fos activity after 2 days of drug-washout, perfusing the mice 11-12h after the time of day of drug administration. We find there is no statistically significant difference between the results of control and drug treatments at this time point. This evidence, together with our finding that c-fos expression in the PL of PCP- and METH-treated mice is still increased 13 and 17 days after the end of drug treatment (assessed during the 5-h window in which they previously received the drug), suggests that drug-induced hyperactivity of PL shows time-of-day dependent fluctuation rather than being present across the entire day/night cycle. However, we do not have evidence of the changes in the activity of these neurons across the full sweep of the day/night cycle. We have included these data in Supplementary Fig. 17c and lines 335 to 341.

We have analyzed food consumption on a small cohort of mice and found that there is a small increase in food consumption in drug-treated mice as compared to controls.

4. The clozapine experiments are interesting, but the mechanism(s) governing its effects – and its interaction with local PV+ neurons - are not clear to me. Do the authors propose that clozapine somehow increases the activity of PV+ neurons? If so, how?

We advanced the hypothesis that clozapine may exert its effect by increasing inhibitory inputs to PL glutamatergic neurons, and cited (Cochran, Kennedy et al. 2003) reporting that clozapine treatment reverses the loss of PV caused by PCP (Discussion, lines 410 and 412). Amitai et al. (Amitai, Kuczenski et al. 2012) report that clozapine rescues the PCP-induced loss of PV and rescues the decrease in GAD67 expression in PV neurons. Thus, clozapine may restore PV neurons' ability to produce GABA, rather than increasing the activity of PV+ neurons. However, the suggestion that impaired GABA production in PV neurons may contribute to PL hyperactivity appears less compelling in light of new evidence showing that c-fos expression in the PL during drug-washout shows time-of-day variation. Therefore, we have removed this hypothesis from the Discussion (lines 409-411).

Can the authors also comment on why they think clozapine did not reverse drug-associated hyperlocomotion (Fig 6g) but activation of PV+ neurons did (Fig 7i)?

While PV activation is a cell type-specific and PL-specific manipulation, systemic injections of clozapine likely affect different types of neurons in multiple areas of the brain. Clozapine is an antagonist of multiple dopamine receptors with different levels of affinity, as well as serotonergic receptors, adrenergic receptors, histamine (H1) receptors, and M1–M5 muscarinic receptors. Thus, the fact that clozapine's effects are widespread and not limited to PL PV neurons may explain why chemogenetics and systemic administration have different effects on locomotion.

5. What relevance do these data have to the neural mechanisms governing the rewarding properties of these drugs? Also, some acknowledgement should be made in the paper that different effects might be expected in a model of drug self-administration vs. the experimenter-administered model utilized here.

At present we do not know whether our data are relevant to the rewarding properties of PCP and METH. Investigation of this point is a high priority for us and something we would like to address. That project requires drug self-administration since, as noted, different effects may be expected for self- vs experimenter-administration of drugs (Stefanski, Ladenheim et al. 1999, Stefanski, Lee et al. 2002, Lecca, Cacciapaglia et al. 2007, Lominac, Sacramento et al. 2012). We have included an acknowledgement of these issues in the Discussion (lines 370 to 375). See also our response to question 1, above.

REFERENCES:

AMITAI, N.; KUCZENSKI, R.; BEHRENS, M. M.; MARKOU, A. Repeated phencyclidine administration alters glutamate release and decreases GABA markers in the prefrontal cortex of rats. **Neuropharmacology**, 62, n. 3, p. 1422-1431, Mar 2012.

COCHRAN, S. M.; KENNEDY, M.; MCKERCHAR, C. E.; STEWARD, L. J. *et al.* Induction of metabolic hypofunction and neurochemical deficits after chronic intermittent exposure to phencyclidine: differential modulation by antipsychotic drugs. **Neuropsychopharmacology**, 28, n. 2, p. 265-275, Feb 2003.

LECCA, D.; CACCIAPAGLIA, F.; VALENTINI, V.; ACQUAS, E. *et al.* Differential neurochemical and behavioral adaptation to cocaine after response contingent and noncontingent exposure in the rat. **Psychopharmacology (Berl)**, 191, n. 3, p. 653-667, Apr 2007.

LOMINAC, K. D.; SACRAMENTO, A. D.; SZUMLINSKI, K. K.; KIPPIN, T. E. Distinct neurochemical adaptations within the nucleus accumbens produced by a history of self-administered vs non-contingently administered intravenous methamphetamine. **Neuropsychopharmacology**, 37, n. 3, p. 707-722, Feb 2012.

PASCOLI, V.; TERRIER, J.; HIVER, A.; LÜSCHER, C. Sufficiency of Mesolimbic Dopamine Neuron Stimulation for the Progression to Addiction. **Neuron**, 88, n. 5, p. 1054-1066, Dec 2 2015.

STEFANSKI, R.; LADENHEIM, B.; LEE, S. H.; CADET, J. L. *et al.* Neuroadaptations in the dopaminergic system after active self-administration but not after passive administration of methamphetamine. **Eur J Pharmacol**, 371, n. 2-3, p. 123-135, Apr 29 1999.

STEFANSKI, R.; LEE, S. H.; YASAR, S.; CADET, J. L. *et al.* Lack of persistent changes in the dopaminergic system of rats withdrawn from methamphetamine self-administration. **Eur J Pharmacol**, 439, n. 1-3, p. 59-68, Mar 29 2002.

Reviewer #3 (Remarks to the Author):

This is an interesting study, in which the authors demonstrated phencyclidine (PCP) and methamphetamine (METH), acting on different targets in the brain, induced the same population of glutamatergic neurons in the medial prefrontal cortex (mPFC) to gain GABA and reduce the expression of the vesicular glutamate transporter (VGLUT), which leads to memory deficits. Additional results show that dopaminergic neurons in the ventral tegmental area (VTA) contributed to the gain of GABA in the mPFC. Drug-induced hyperactivity in the mPFC changed the transmitter identity, which can be reversed and repair the memory deficits. Overall, their experimental design is coherent, the data appear to be solid, the conclusions are reasonable, and the manuscript is well written. Below are a few minor points for the authors to consider.

1. The mPFC is a major hub for cognitive control (Line 56), and the study focused on the prelimbic subregion (PL). However, the mPFC consists of several areas including both the prelimbic and infralimbic cortex that are involved in different cognitive performances. An explanation would be helpful for readers to understand the selection of brain regions.

We focused on the PL, rather than the infralimbic or anterior cingulate cortex, because preliminary examination of these latter two subregions did not reveal gain of GABA in response to PCP administration.

The prelimbic subregion of the mPFC has been shown to regulate cognitive function in mice, including both recognition memory assessed by the novel object recognition test and working memory assessed by the spontaneous alternation task (Divac, Wikmark et al. 1975, Kamei, Nagai et al. 2006). The infralimbic cortex and the anterior cingulate cortex also regulate cognitive functions (Cassaday, Nelson et al. 2014, Diehl and Redish 2023).

2. In the supplementary Fig.10, the authors performed chemogenetic inhibition of VTA dopaminergic neurons, then used c-fos expression to examine the effect of virus. To detect the change of excitability in VTA dopaminergic neurons, action potential firing is a more direct indicator.

Action potential firing is indeed a more direct indicator. We have established a collaboration with a superb electrophysiologist in Professor Byungkook Lim's lab who has been able to carry out the experiments addressing points 3 and 4, below. Unfortunately, we have not been able to coordinate chemogenetic inhibition of VTA neurons with intracellular recording in order to address this point 2.

3. In the supplementary Fig.11, to test the effectiveness of YFP-ChR2, it's necessary to verify the optogenetic stimulation in the brain slices using electrophysiological approach.

We have performed experiments coordinating optogenetic stimulation in brain slices with intracellular recordings. The results confirm the success of optogenetic stimulation and are now included in new Supplementary Fig. 13a,b.

4. In the supplementary Fig.12, both PCP and METH increased c-fos expression in PL glutamatergic neurons. To confirm that chronic drug treatment increases the excitability of PL glutamatergic neurons, spontaneous and evoked action potential recordings are preferred.

We have now treated mice with a 10-day administration of PCP, METH or saline and recorded from PL glutamatergic neurons. The results of these experiments are included here. We predicted the existence of two populations of neurons: one in which excitability was increased (c-fos-positive neurons) and one in which excitability was not increased (c-fos-negative neurons). However, we saw no evidence of two populations of neurons. Because only 10% of mCherry+ glutamatergic neurons express c-fos in response to these drugs, only 1 in 10 recordings would be expected to show changes in excitability. Since recordings were made from brain slices of living tissue, we could not identify the cells that were expressing c-fos at the time of the recordings. As a result, there was no way to match the recordings to the presence or absence of

c-fos. We regret that it was not possible to establish, in this system, a correlation of *c-fos* staining with the presence of increased excitability.

No changes in the excitability of PL glutamatergic neurons were detected in PCP- and METH-treated mice compared to controls 2 days after the end of drug treatment. **a.** Experimental protocol. **b.** Representative traces of PL pyramidal neuron firing in response to 300 pA of injected current. **c.** Changes in the firing rate of PL pyramidal neurons as a function of injected current (n=3 mice). **d.** Comparison of rheobase across treatment groups shows no statistically significant differences (assessed using one-way ANOVA with Dunnett's multiple-comparisons test). Recordings were performed from pyramidal neurons in layer 2/3 of the PL, using the procedure described in lines 721 to 737 of the Methods.

5. PCP mainly interacts with NMDA receptor, and METH mainly interact with dopamine transporter. To help understand the mechanism, a discussion is needed about how PCP and METH induce similar effects on the excitability of VTA dopaminergic neurons and PL glutamatergic neurons, and the roles of VTA-mPFC circuit in regulating cognitive behaviors.

PCP and METH, like virtually all other addictive substances, increase DA release in the mesocorticolimbic system, mimicking the effect of phasic firing of dopaminergic neurons in the VTA (Mele, Wozniak et al. 1998, Volkow, Michaelides et al. 2019). As the VTA and the PL are reciprocally connected, increased release of dopamine from VTA neurons can influence the activity of the PL (Lohani, Martig et al. 2019). Furthermore, increased dopamine release in regions other than the PL can also affect the firing of PL neurons indirectly via multi-synaptic pathways (Hsieh, Stein et al. 2014).

PCP is an NMDA receptor antagonist that shows affinity for the dopamine transporter only at high anesthetic doses. Thus, PCP-induced increase in mesocorticolimbic dopamine release does not depend on PCP's direct effect on dopaminergic neurons, but appears to be mediated indirectly by PCP's ability to influence glutamatergic transmission (Adams and Moghaddam 1998) and increase firing rate of VTA DA neurons (Freeman and Bunney 1984; French 1994). Methamphetamine, instead, acts directly on dopaminergic cells to promote the release of

dopamine (Barr, Panenka et al. 2006). We have clarified this point in the Discussion (lines 377 to 379).

REFERENCES:

ADAMS, B.; MOGHADDAM, B. Corticolimbic Dopamine Neurotransmission Is Temporally Dissociated from the Cognitive and Locomotor Effects of Phencyclidine. **The Journal of Neuroscience**, 18, n. 14, p. 5545-5554, 1998.

BARR, A. M.; PANENKA, W. J.; MACEWAN, G. W.; THORNTON, A. E. *et al.* The need for speed: an update on methamphetamine addiction. **J Psychiatry Neurosci**, 31, n. 5, p. 301-313, Sep 2006.

CASSADAY, H. J.; NELSON, A. J.; PEZZE, M. A. From attention to memory along the dorsal-ventral axis of the medial prefrontal cortex: some methodological considerations. **Front Syst Neurosci**, 8, p. 160, 2014.

DIEHL, G. W.; REDISH, A. D. Differential processing of decision information in subregions of rodent medial prefrontal cortex. **Elife**, 12, Jan 18 2023.

DIVAC, I.; WIKMARK, R. G. E.; GADE, A. Spontaneous alternation in rats with lesions in the frontal lobes: An extension of the frontal lobe syndrome. **Physiological Psychology**, 3, n. 1, p. 39-42, 1975/03/01 1975.

FREEMAN, A. S.; BUNNEY, B. S. The effects of phencyclidine and N-allylnormetazocine on midbrain dopamine neuronal activity. **Eur J Pharmacol**, 104, n. 3-4, p. 287-293, Sep 17 1984.

FRENCH, E. D. Phencyclidine and the midbrain dopamine system: Electrophysiology and behavior. **Neurotoxicology and Teratology**, 16, n. 4, p. 355-362, 1994/07/01/ 1994.

HSIEH, J. H.; STEIN, D. J.; HOWELLS, F. M. The neurobiology of methamphetamine induced psychosis. **Front Hum Neurosci**, 8, p. 537, 2014.

KAMEI, H.; NAGAI, T.; NAKANO, H.; TOGAN, Y. *et al.* Repeated methamphetamine treatment impairs recognition memory through a failure of novelty-induced ERK1/2 activation in the prefrontal cortex of mice. **Biol Psychiatry**, 59, n. 1, p. 75-84, Jan 1 2006.

LOHANI, S.; MARTIG, A. K.; DEISSEROTH, K.; WITTEN, I. B. *et al.* Dopamine Modulation of Prefrontal Cortex Activity Is Manifold and Operates at Multiple Temporal and Spatial Scales. **Cell Rep**, 27, n. 1, p. 99-114.e116, Apr 2 2019.

MELE, A.; WOZNIAK, K. M.; HALL, F. S.; PERT, A. The role of striatal dopaminergic mechanisms in rotational behavior induced by phencyclidine and phencyclidine-like drugs. **Psychopharmacology (Berl)**, 135, n. 2, p. 107-118, Jan 1998.

VOLKOW, N. D.; MICHAELIDES, M.; BALER, R. The Neuroscience of Drug Reward and Addiction. **Physiol Rev**, 99, n. 4, p. 2115-2140, Oct 1 2019.

REVIEWERS' COMMENTS

Reviewer #1 (Remarks to the Author):

The authors have responded to the concerns of this reviewer and the resulting manuscript has improved, with a more clear explanation of the putative mechanisms mediating this interesting phenomenon.

Reviewer #2 (Remarks to the Author):

The authors have been extremely responsive to my previous comments. I especially appreciate the additional work required to carry out the PL terminal stimulation and time-of-day experiments. The new data raise interesting additional questions that will require extensive experimentation; thus these experiments are beyond the scope of the current manuscript. The authors have done a nice job of speculating about what these new data could mean in the Discussion section. In my view, this manuscript is now suitable for publication at the journal. Congratulations on a very interesting paper.
- Morgan James

Reviewer #3 (Remarks to the Author):

The authors did a great job in addressing previous concerns and revising the manuscript. Some electrophysiological data cannot support the findings under the current conditions, but the explanation is reasonable. In addition, there is a mistake in the rebuttal letter (Reviewer #3, Point 3), and I think the authors meant the new Supplementary Fig. 12a,b (not Fig. 13a,b). There are no additional questions about this manuscript.